# ERAP1 promotes Hedgehog-dependent tumorigenesis by controlling USP47-mediated degradation of βTrCP

Francesca Bufalieri [1,17], Paola Infante [2,17], Flavia Bernardi [1,3,4], Miriam Caimano [1], Paolo Romania [5], Marta Moretti [1], Ludovica Lospinoso Severini [1,3,4], Julie Talbot [3,4], Ombretta Melaiu [5,6], Mirella Tanori [7], Laura Di Magno [2], Diana Bellavia [1], Carlo Capalbo [1], Stéphanie Puget [8], Enrico De Smaele [9], Gianluca Canettieri [1,10], Daniele Guardavaccaro [11], Luca Busino [12], Angelo Peschiaroli [13], Simonetta Pazzaglia [7], Giuseppe Giannini [1,10], Gerry Melino [14,15], Franco Locatelli [5,16], Alberto Gulino [1], Olivier Ayrault [3,4], Doriana Fruci [5] & Lucia Di Marcotullio [1,10]

The Hedgehog (Hh) pathway is essential for embryonic development and tissue homeostasis. Aberrant Hh signaling may occur in a wide range of human cancers, such as medulloblastoma, the most common brain malignancy in childhood. Here, we identify endoplasmic reticulum aminopeptidase 1 (ERAP1), a key regulator of innate and adaptive antitumor immune responses, as a previously unknown player in the Hh signaling pathway. We demonstrate that ERAP1 binds the deubiquitylase enzyme USP47, displaces the USP47-associated βTrCP, the substrate-receptor subunit of the SCF$^{βTrCP}$ ubiquitin ligase, and promotes βTrCP degradation. These events result in the modulation of Gli transcription factors, the final effectors of the Hh pathway, and the enhancement of Hh activity. Remarkably, genetic or pharmacological inhibition of ERAP1 suppresses Hh-dependent tumor growth in vitro and in vivo. Our findings unveil an unexpected role for ERAP1 in cancer and indicate ERAP1 as a promising therapeutic target for Hh-driven tumors.

[1] Department of Molecular Medicine, University La Sapienza, 00161 Rome, Italy. [2] Center for Life NanoScience@Sapienza, Istituto Italiano di Tecnologia, 00161 Rome, Italy. [3] Institut Curie, PSL Research University, CNRS UMR, INSERM, 91405 Orsay, France. [4] Université Paris Sud, Université Paris-Saclay, CNRS UMR 3347, INSERM U1021 Orsay, France. [5] Paediatric Haematology/Oncology Department, Ospedale Pediatrico Bambino Gesù, IRCCS, 00146 Rome, Italy. [6] Department of Biology, University of Pisa, 56126 Pisa, Italy. [7] Laboratory of Biomedical Technologies, Agenzia Nazionale per le Nuove Tecnologie, l'Energia e lo Sviluppo Economico Sostenibile (ENEA), S.Maria di Galeria, 00123 Rome, Italy. [8] Department of Pediatric Neurosurgery, Necker University Hospital, University Paris Descartes, Sorbonne Paris Cité, 75015 Paris, France. [9] Department of Experimental Medicine, University La Sapienza, 00161 Rome, Italy. [10] Istituto Pasteur-Fondazione Cenci Bolognetti, University La Sapienza, 00161 Rome, Italy. [11] Department of Biotechnology, University of Verona, 37134 Verona, Italy. [12] Department of Cancer Biology, Perelman School of Medicine, University of Pennsylvania, Philadelphia 19104 PA, USA. [13] National Research Council of Italy CNR, Institute of Translational Pharmacology (IFT), 00133 Rome, Italy. [14] MRC Toxicology Unit, University of Cambridge, LE17HB Leiicester, UK. [15] Dipartimento Medicina Sperimentale e Chirurgia, Università Tor Vergata, 00133 Rome, Italy. [16] Department of Pediatrics, University La Sapienza, 00161 Rome, Italy. [17]These authors contributed equally: Francesca Bufalieri, Paola Infante. Correspondence and requests for materials should be addressed to Doriana Fruci (email: doriana.fruci@opbg.net) or to L. Di Marcotullio (email: lucia.dimarcotullio@uniroma1.it)

The Hedgehog (Hh) signaling pathway plays a crucial role during organogenesis and stem cells maintenance. Its deregulation is responsible for the onset of several human cancers[1–5]. Hh signaling is triggered by the binding of the Hh ligand to the Patched1 (Ptch1) receptor, relieving the repression on the Smoothened (Smo) co-receptor. This event leads activation of the Gli family transcription factors upon the dissociation from SuFu protein, an important negative regulator of the pathway[2,6]. In mammals, three Gli proteins have been identified: Gli1 and Gli2 with activating functions and Gli3 mainly working as repressor[7]. Ubiquitin-dependent proteolytic processing of Gli transcription factors is important for controlling the Hh pathway output. Of note, all Gli transcription factors undergo ubiquitination through β-transducin-repeat containing E3 ubiquitin protein ligase (βTrCP), an F-box protein of the Skp1–Cul1–Fbox protein (SCF) E3-ligase complex, which promotes the complete degradation of Gli1 and Gli2[8–11], and the processing of Gli3 into the repressor form Gli3R[12–14]. Deregulation of these events results in uncontrolled cell proliferation and tumorigenesis[9]. One of the most relevant Hh-dependent tumors is medulloblastoma (MB), a highly aggressive pediatric malignancy arising from cerebellar granule cell progenitors (GCPs) mainly mutated in Ptch1 or Smo receptors[15]. The non-canonical Hh/Gli activation regardless of the presence of mutated components of the pathway or overexpression of the ligand is also frequently observed in MB and other tumors. This emphasizes the importance of the mechanisms controlling Gli activity, which are impaired in disease[16,17]. Pharmacological inhibition of the Hh pathway has been proposed as a therapeutic strategy in typical Hh-dependent, tumors such as advanced basal cell carcinoma (BCC) and medulloblastoma (MB)[18,19]. Hence, to identify novel Hh antagonists we tested a number of small molecules and found leucinethiol (Leu-SH), an inhibitor of Endoplasmic reticulum aminopeptidase 1 (ERAP1)[20–28], as one of Hh inhibitors.

ERAP1 is crucial for the maturation of a wide spectrum of substrates involved in multiple biological processes, including antigen processing and regulation of blood pressure and inflammation[29]. Functional single nucleotide polymorphisms in ERAP1 have been associated with predisposition to several human diseases, including autoimmune diseases, viral infections and virally-induced cancer[30]. Reduction of ERAP1 expression by RNA interference results in a drastic change in the repertoire of antigenic peptides presented by MHC class I molecules[21,31,32]. In mouse models, the complete loss of ERAAP expression, the mouse homologous of ERAP1, inhibits surface expression of MHC class I molecules. In human neoplastic lesions, the expression of ERAP1 differs as compared to the normal counterparts, depending on the tumor type[33]. In general, loss of ERAP1 is frequently associated with the lack of detectable MHC class I surface expression, potentially contributing to tumor immunoescape[34]. Consistently, in cervical carcinoma, altered ERAP1 expression is linked to poor clinical outcome, suggesting that an aberrant antigen processing may contribute to escape the host immune surveillance[35]. Accordingly, inhibition of ERAP1 was shown to generate strong innate and adaptive anti-tumor immune responses resulting in tumor regression in two distinct tumor mouse models, thus providing evidences that modulation of ERAP1 activity may represent a promising tool for cancer immunotherapy[27,36].

Here, we identified a role of ERAP1 in tumorigenesis, acting through the activation of the Hh pathway. We demonstrate that ERAP1 induces the degradation of βTrCP by physically interacting with βTrCP-bound deubiquitylase enzyme USP47. This event protects Gli transcription factors from βTrCP-dependent degradation and stimulates Hh activity. Remarkably, both genetic and pharmacological inhibition of ERAP1 suppresses Hh-dependent tumor growth in vitro and in vivo, suggesting an innovative therapeutic approach in the treatment of Hh-dependent tumors.

## Results

**ERAP1 positively regulates the Hh pathway.** While exploring novel Hh antagonists we found Leu-SH[27,28], an inhibitor of ERAP1[20–28], as one of Hh inhibitors. Leu-SH significantly reduced the luciferase activity of the Hh pathway-reporter in NIH Shh-Light II cells activated via SAG treatment[37] (Fig. 1a) but not the one driven by Hh-unrelated or Hh-related signaling pathway (i.e. Jun/AP1 and Wnt/β-Catenin, respectively) (Supplementary Fig. 1). To explore the role of ERAP1 in the regulation of the Hh pathway, we inhibited genetically or pharmacologically ERAP1 in in vitro models. Following silencing of the murine ER aminopeptidase ERAAP (herein referred as ERAP1) by means of short hairpin RNAs and treatment with the Hh pathway agonist SAG[38], we observed a reduced expression of Gli1, the final and most powerful effector of Hh signaling, both at mRNA and protein levels (Fig. 1b, c). Consistent with these data, treatment with Leu-SH of $Ptch^{−/−}$ mouse embryonic fibroblasts (MEFs), in which the pathway is constitutively active[39], induced a significant reduction in the expression of several endogenous Hh target genes (Fig. 1d, e). To investigate if ERAP1 is acting at post-receptor level, we used $SuFu^{−/−}$ MEFs, in which the pathway is active due to the loss of the well-known Gli inhibitor SuFu[40]. Leu-SH treatment determined a downregulation of the Hh signature gene also in this cellular context (Fig. 1f). Similar results were achieved following genetic ERAP1 depletion in both cell models (Fig. 1g, h), indicating that ERAP1 promotes the Hh signaling acting downstream of SuFu.

**ERAP1 activates Hh signaling by impairing βTrCP stability.** To better characterize the molecular mechanism whereby ERAP1 regulates the Hh pathway, we analyzed the protein levels of Gli transcription factors in MEFs upon expression of increasing amounts of ERAP1 (Fig. 2a). We observed that while Gli1 and Gli2 were increased, both the full length and the cleaved form of the Gli3 protein (Gli3FL and Gli3R, respectively) were reduced (Fig. 2a). The opposite effect was observed following treatment with increasing amounts of Leu-SH (Fig. 2b), which impairs the enzymatic activity of ERAP1 without affecting its protein level (Supplementary Fig. 2a). Of note, Leu-SH treatment was able to antagonize the activation of Hh signaling induced by SAG, leading to decreased expression of Gli1 and Gli2 and increased levels of Gli3FL and Gli3R (Fig. 2c). Since ERAP1 did not directly associate with Gli transcription factors (Supplementary Fig. 2b), we hypothesized that ERAP1 could regulate βTrCP, an F-box protein belonging to a Skp1/Cul1/F-box E3 ubiquitin ligase complex crucial for the degradation of Gli1[9] and Gli2[8,10], and the formation of Gli3R[14,41,42]. Dose-dependent overexpression of ERAP1 led to reduction of βTrCP protein levels (Fig. 2d), whereas its pharmacological or genetic inhibition resulted in the opposite effect (Fig. 2e, f). Importantly, the reintroduction of ERAP1 in ERAP1-knockdown cells downregulated βTrCP leading to increased Gli1, to a lesser extend and Gli2 and decreased Gli3FL and Gli3R levels (Fig. 2f). No change in βTrCP mRNA expression was detected in both assays (Supplementary Fig. 3a, b). Importantly, the modulation of ERAP1 had no effect on other E3 ligases involved in Gli ubiquitylation, such as Itch[43,44], pCAF[45], SPOP[46] or SCF ubiquitin ligase Skp2[47] (Fig. 2d–f), as well as on other βTrCP substrates[48] (Supplementary Fig. 3c,d). Notably, ERAP1 did not affect Gli1 protein levels in the absence of βTrCP, indicating that the latter is required for ERAP1-mediated regulation of the Hh pathway (Fig. 2g). Moreover, βTrCP half-life was

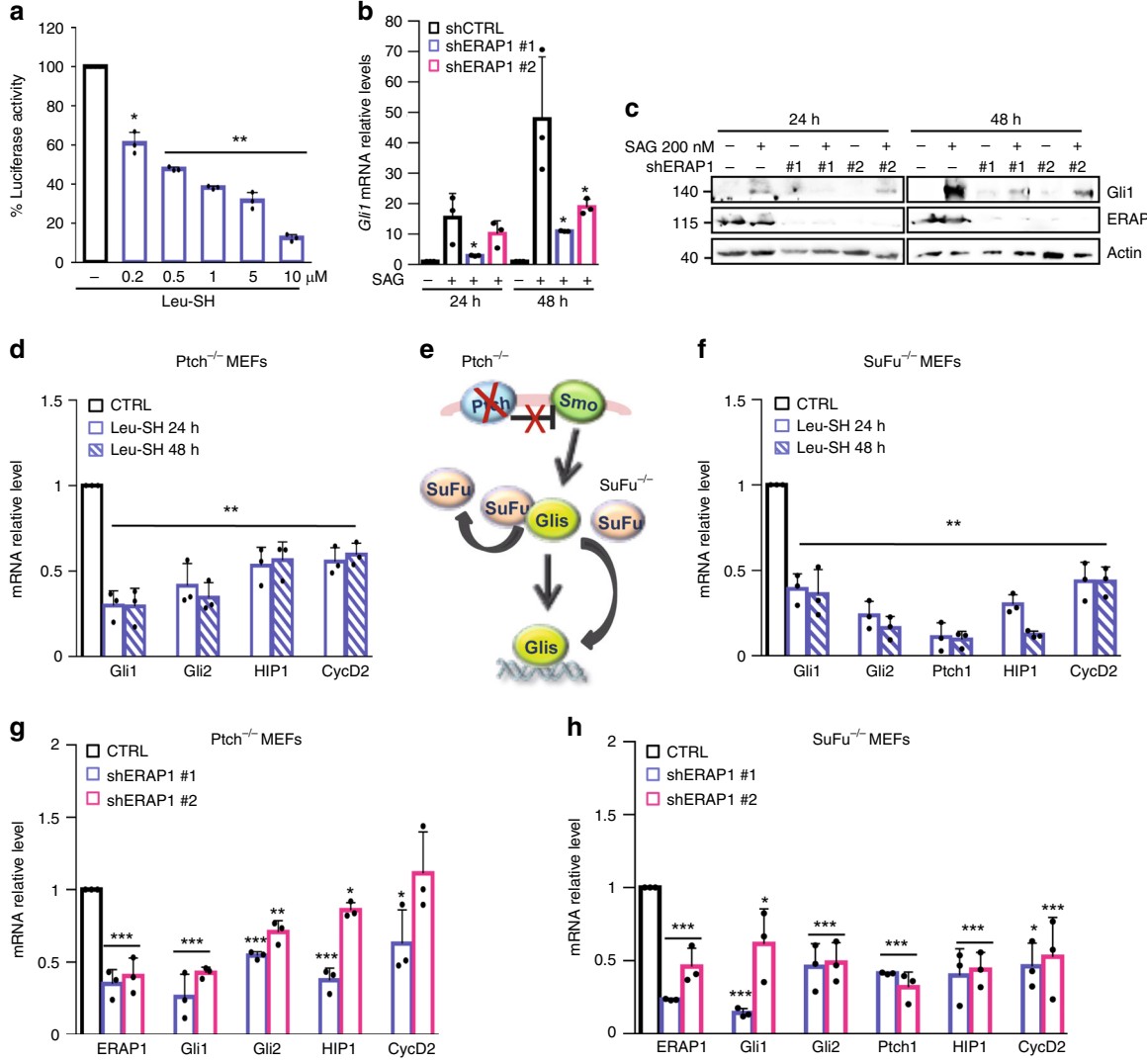

**Fig. 1** ERAP1 positively regulates the Hh pathway at postreceptor level. **a** Luciferase activity of NIH3T3 Shh-Light II cells treated for 24 h with SAG and increasing amounts of Leu-SH or DTT as control. **b**, **c** Quantitative real-time PCR (qRT-PCR) (**b**) and representative immunoblotting (**c**) analyses of Gli1 expression in the NIH3T3 murine fibroblasts transduced with lentiviral vectors encoding either control shRNA (shCTRL) or ERAP1 shRNA (shERAP1#1 and shERAP1#2) and treated with SAG or DMSO for either 24 or 48 h. In **c** ERAP1 expression was also evaluated and actin was used as loading control. **d**, **f** qRT-PCR analysis of Hh target genes expression in Ptch$^{-/-}$ (**d**) and SuFu$^{-/-}$ MEFs (**f**) both treated with Leu-SH (30 μM) or DTT as control. **e** Representative model of the constitutive activation of Smo or Gli1 in Ptch$^{-/-}$ and SuFu$^{-/-}$ MEFs, respectively. **g**, **h** qRT-PCR analysis of Hh target genes expression in Ptch$^{-/-}$ (**g**) and SuFu$^{-/-}$ MEFs (**h**) transduced with shCTRL or shERAP1 constructs. Data in **b**, **d**, **f**, **g**, and **h** are normalized to endogenous GAPDH and HPRT controls and expressed as the fold change respect to the control sample value. All data represent the mean of three independent experiments. Mean ± SD; *$P < 0.05$; **$P < 0.01$; ***$P < 0.001$ calculated with two-sided Student's t-test

shorter following ERAP1 overexpression (Fig. 2h), suggesting that ERAP1 controls βTrCP protein stability.

Next, we evaluated whether ERAP1 controls βTrCP levels by modulating its ubiquitylation. To this purpose, we performed an in vivo ubiquitylation assay upon ectopic expression of ERAP1. High levels of ERAP1 promoted the poly-ubiquitylation of endogenous βTrCP and its subsequent degradation by the proteasome, leading to a significant accumulation of βTrCP protein and its ubiquitylated forms in cells treated with the proteasome-inhibitor MG132 (Fig. 2i). Conversely, βTrCP poly-ubiquitylation was decreased in MEFs treated with increasing amount of Leu-SH (Fig. 2j). As the phosphorylation of the Gli transcription factors within the DSG(X)$_{2+n}$S destruction motifs (degron) is required for the interaction of βTrCP[9,41,49], we evaluated if the effect of ERAP1 depended upon the presence of

this motif. To this purpose, MEFs were transfected with the wild type (HA-Gli1 WT) or the mutant (HA-Gli1ΔC) form of Gli1 lacking the degron essential for βTrCP recognition[9], and then treated with increasing amount of Leu-SH. Inhibition of ERAP1 induced a reduction in the levels of the wild type, but not of the mutated form of Gli1 (Fig. 2k). The same result was obtained in MEFs transduced with shCTRL or shERAP1 (Fig. 2l). Interestingly, we also observed that Gli1 is phosphorylated in Ptch$^{-/-}$ MEFs by Phospho-Kinase A (PKA) (Supplementary Figure 4), the kinase that triggers the phosphorylation cascade of Gli transcription factors required for βTrCP recognition, being increased after treatment with PKA inhibitor[50,51].

Altogether these data demonstrate that ERAP1 promotes Hh pathway activity by impairing βTrCP protein levels leading to the modulation of Gli proteins.

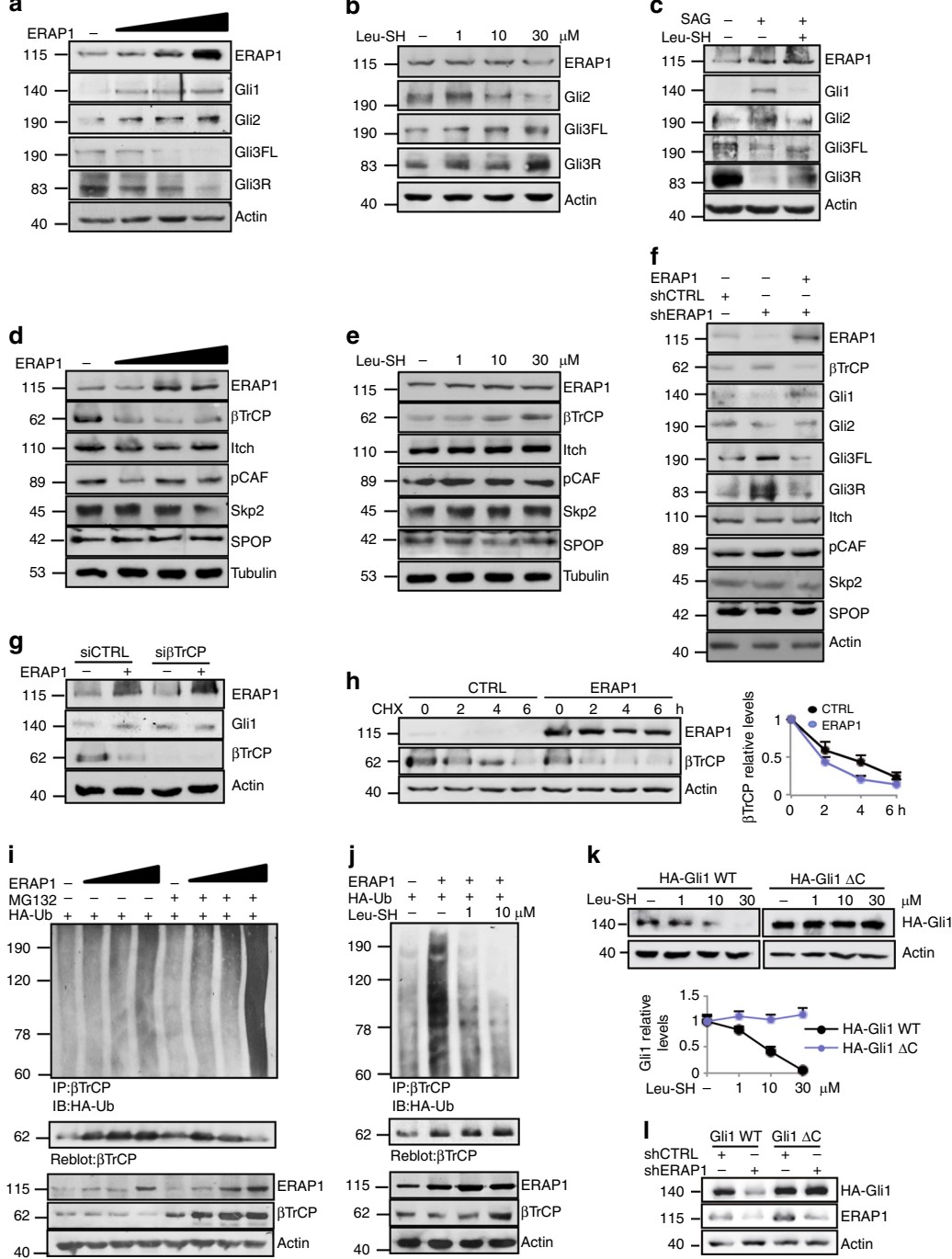

**Fig. 2** ERAP1 activates Hh signaling by impairing βTrCP protein expression. **a–f** Representative immunoblotting analyses of the indicated proteins in MEFs transfected with increasing amounts of vector encoding ERAP1 (**a**, **d**) or treated for 24 h with Leu-SH at the indicated concentration (**b**, **e**), or SAG (200 nM) and Leu-SH (30 μM) (**c**), or DTT as control. In **f** MEFs were transduced with shCTRL or shERAP1 and transfected with a vector encoding ERAP1. Actin (**a–c**, **f**) and tubulin (**d**, **e**) were used as loading controls. **g** ERAP1, Gli1 and βTrCP protein levels in MEFs transfected with an empty vector or a vector encoding ERAP1 in the presence of small interfering RNAs (siRNAs) to a non-relevant mRNA (siCTRL) or murine βTrCP mRNA (siβTrCP). **h** βTrCP protein levels in MEFs transfected with an empty vector or a vector encoding ERAP1 and treated with cycloheximide (CHX, 100 μg/mL) at different time points. Densitometry analysis of actin-normalized βTrCP values of three independent experiments is shown (right panel). **i**, **j** Endogenous βTrCP was immunoprecipitated from MEFs expressing the indicated proteins and treated with MG132 (50 μM) for 4 h (**i**) or increasing doses of Leu-SH for 24 h (**j**), followed by immunoblotting with an anti-HA antibody to detect conjugated HA-Ub. Blots were both reprobed with a βTrCP antibody. Bottom ERAP1 and βTrCP protein levels in total cell lysate. Actin was used as loading control. **k** Immunoblotting (upper panel) and densitometric analysis (lower panel) of HA-Gli1 WT or HA-Gli1ΔC protein levels transfected in MEFs and treated after 24 h with increasing amount of Leu-SH for 24 h. **l** Immunoblotting analysis of HA-Gli1 WT or HA-Gli1ΔC protein levels transfected in MEFs transduced with shCTRL or shERAP1. Actin was used as loading control

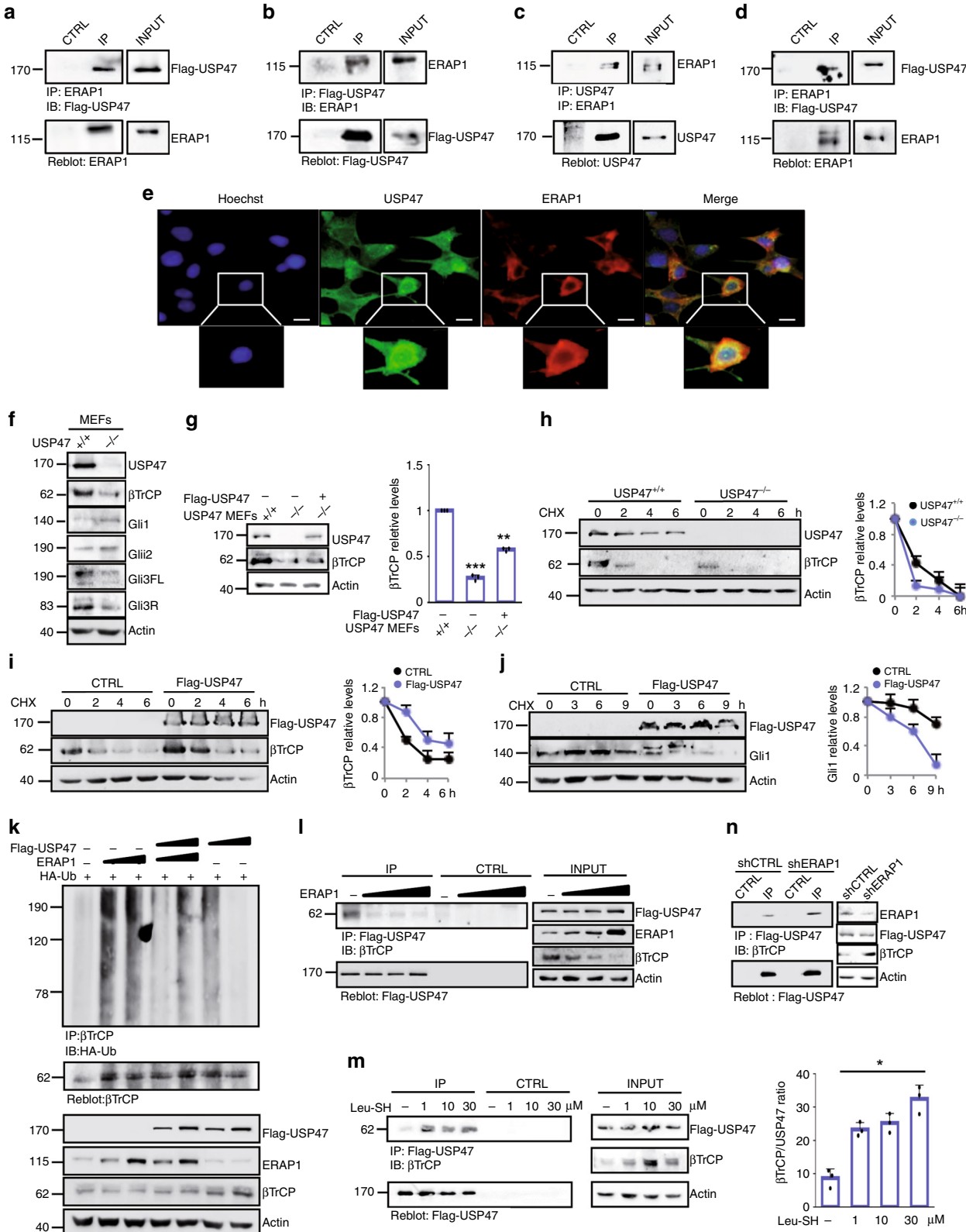

**ERAP1 promotes βTrCP degradation by interacting with USP47.** As ERAP1 did not appear to interact directly with βTrCP (Supplementary Figs. 5a, b), we investigated whether ERAP1 interferes with the Ubiquitin-Specific Protease 47 (USP47), a deubiquitylating enzyme known to interact with βTrCP[52]. Co-immunoprecipitation experiments demonstrated that ERAP1 binds both exogenous and endogenous USP47 in MEFs

(Fig. 3a–d). Accordingly, confocal microscopy revealed that ERAP1 co-localizes with endogenous USP47 proteins in the perinuclear region, where endogenous ERAP1 is mainly localized[31] (Fig. 3e). Further, ERAP1 did not bind other Ubiquitin-Specific Proteases known to regulate the stability or activity of βTrCP, such as USP24[53] and USP22[54] (Supplementary Fig. 5c). βTrCP stability was impaired in USP47−/− MEFs, leading to

**Fig. 3** ERAP1 promotes βTrCP ubiquitylation by interacting with USP47. **a–d** MEFs were transfected with ERAP1 and/or Flag-USP47. Interaction between USP47 and ERAP1 was detected by immunoprecipitation followed by immunoblot analysis with the indicated antibodies. **e** MEFs transfected with ERAP1 were stained with anti-ERAP1 and anti-USP47 antibodies. Green and red, USP47 and ERAP1 expressing cells, respectively. Nuclei were counter stained with Hoechst (Blue). Magnification ×60; Bars: 5 μm. Representative images from three independent experiments. **f** βTrCP and Gli steady state in USP47$^{+/+}$ and USP47$^{-/-}$ MEFs. **g** βTrCP protein level in USP47$^{+/+}$, USP47$^{-/-}$ and USP47$^{-/-}$ Flag-USP47 transfected MEFs. **h** βTrCP half-life in USP47$^{+/+}$ vs. USP47$^{-/-}$ MEFs treated with CHX (100 μg/mL) at the indicated times. **i** βTrCP protein levels in MEFs transfected with empty vector as control or Flag-USP47 and treated with CHX (100 μg/mL) at different time points. **j** Gli1 protein levels in Ptch$^{-/-}$ MEFs transfected with empty vector as control or Flag-USP47 and treated after 24 h with CHX (100 μg/mL) for different time points. In **g–j** densitometric analysis of βTrCP and Gli1 protein levels of three independent experiments are shown (right panels). **k** MEFs were transfected with HA-Ub and increasing amount of ERAP1 in the presence or absence of Flag-USP47. Endogenous βTrCP was immunoprecipitated with an anti-βTrCP antibody and the ubiquitylated forms were revealed with an anti-HA antibody (upper panel). The blot was reprobed with an anti-βTrCP antibody. Flag-USP47, ERAP1 and βTrCP total protein levels are shown (lower panel). **l** MEFs were transfected with Flag-USP47 and increasing amount of ERAP1. Interaction between Flag-USP47 and endogenous βTrCP was assessed by immunoprecipitation and immunoblotting with the indicated antibodies. Actin was used as loading control. **m** MEFs were transfected with Flag-USP47 and treated for 24 h with Leu-SH at the indicated concentration. Interaction between Flag-USP47 and endogenous βTrCP was detected as described in **l**. Densitometric analysis of the Flag-USP47/βTrCP binding ratio representative of three independent experiments is shown (right panel). **n** MEFs were transduced with shCTRL or shERAP1 and transfected with Flag-USP47. Interaction between Flag-USP47 and endogenous βTrCP was assessed as described in **l**

increased expression of Gli1 and Gli2 and decreased expression of Gli3FL and Gli3R (Fig. 3f). Consistently, the re-introduction of USP47 in USP47$^{-/-}$ MEFs restored βTrCP protein levels (Fig. 3g). Moreover, βTrCP half-life was reduced in the USP47$^{-/-}$ MEFs (Fig. 3h) and increased in the presence of USP47 as compared to control cells (Fig. 3i). Of note, USP47 overexpression decreased Gli1 half-life (Fig. 3j) in agreement with the established role of βTrCP in Gli1 regulation.

The effect of USP47 on ERAP1-mediated βTrCP ubiquitylation was studied by performing an in vivo βTrCP ubiquitylation assay in the presence of ectopic expression of ERAP1 and/or USP47. High levels of ERAP1 promoted a robust ubiquitylation of βTrCP that was counteracted by the co-expression of USP47 (Fig. 3k). Accordingly, the overexpression of USP47 alone leads to a decrease of ubiquitylated βTrCP (Fig. 3k) consistent with the enzymatic function of USP47[52]. Importantly, ERAP1 strongly impaired βTrCP/USP47 interaction (Fig. 3l), whereas its pharmacological inhibition or genetic depletion resulted in an increased association between the two proteins (Fig. 3m, n).

Overall, these findings indicate that ERAP1 promotes ubiquitylation and degradation of βTrCP by displacing its interaction with the USP47 deubiquitylase enzyme.

**ERAP1 affects Hh-dependent growth of cerebellar GCPs.** Hh signaling crucially regulates cerebellar development by controlling the expansion of a subset of granule cell progenitors (GCPs) and the proper development of the granule neuron lineage under Purkinje cell-derived Shh stimuli. Withdrawal of Hh signal causes physiologically GCPs growth arrest after the first post-natal week in mice inducing their differentiation into mature granules[55]. Importantly, genetic or epigenetic alterations in the Hh signaling lead to GCPs increased proliferation and their tumorigenic conversion[56,57]. To investigate the biological role of ERAP1 on Hh-dependent growth, the levels of ERAP1 expression were analyzed in GCPs at an early post-natal stage that is Hh-dependent. Similar to Gli1, ERAP1 was mainly expressed in the Hh-dependent outer external germinal layer (EGL) where highly proliferating GCPs reside, and absent in non-proliferating inner germinal layer (IGL) GCPs (Fig. 4a). Consistently, the proliferation rate and Gli1 expression levels of SAG-treated GCPs were significantly reduced in the presence of high doses of Leu-SH (Fig. 4b, c) or following the genetic inhibition of ERAP1 (Fig. 4d–f). As expected, the opposite effect was observed in SAG-treated GCPs overexpressing ERAP1 (Fig. 4g–i), suggesting a potential role of ERAP1 in controlling GCPs proliferation through Hh signaling.

**ERAP1 affects Hh-dependent tumor cell growth in vitro.** GCPs are considered the cells of origin of MB, the most common pediatric brain tumor genetically classified in four subgroups, of which Shh-group is the best characterized[1,58]. The relevance of ERAP1 on Hh-dependent tumor cell growth was determined by testing short-term cultures of primary MB cells freshly isolated from Math1-cre/Ptc$^{C/C}$ mice tumors, one the most used model to study the Hh-dependent tumorigenesis[39,57,59–61]. Pharmacological inhibition of ERAP1 significantly reduced the proliferation of Math1-cre/Ptc$^{C/C}$ MB cells in a dose- and time-dependent manner (Fig. 5a). This was consistent with increased cell death (Fig. 5b), increased levels of the cleaved Caspase 3 protein (Fig. 5c) and reduced BrdU uptake (Fig. 5d). Accordingly, Gli1 expression was reduced at mRNA and protein levels (Fig. 5e, f), indicating an impairment of the Hh signaling activity. Given the difficulty to obtain stable Hh-dependent MB cell lines, tumor cells from spontaneous MB of Math1-cre/Ptc$^{C/C}$ mice were propagated as neurospheres (MB Stem-Like Cells, MB-SLCs) in EGF- and bFGF-free cultured medium to retain the characteristic of in vivo Hh-subtype MB and preventing the differentiation of GCPs[62]. Similarly to cerebellar progenitors, the pharmacological inhibition of ERAP1 impaired both tumor cell proliferation in a dose- and time-dependent manner (Fig. 5g) and clonogenic self-renewal ability (Fig. 5h). Accordingly, MB neurospheres treated with increasing amounts of Leu-SH showed an impaired Hh pathway activity as evaluated by the significant reduced expression of the Hh pathway target genes Gli1, Gli2, and Ptch1, stemness markers (Oct4 and Nanog) and growth (CycD2) and oncogenic (N-Myc) related signals (Fig. 5i, j). Overall, these data demonstrate that ERAP1 affects Hh-dependent MB cell proliferation.

**ERAP1 affects Hh-dependent tumor growth in vivo.** Based on the in vitro studies, we hypothesized that inhibition of ERAP1 activity may reduce tumor growth in vivo. To address this critical issue NOD/SCID gamma (NSG) mice were grafted with spontaneous primary MB from Math1-cre/Ptc$^{C/C}$ mice and the obtained tumor masses were treated with Leu-SH or vehicle for about three weeks[27]. Leu-SH treatment significantly reduced tumor growth as compared to controls (Fig. 6a, b). Reduced cellularity associated with a decreased Ki67 and increased NeuN and cleaved Caspase-3 positive tumor cells was detected in Leu-SH treated tumor masses, indicating that ERAP1 inhibition impairs tumor growth by promoting cell differentiation and committing tumor cells to apoptosis (Fig. 6c, d). In agreement with the in vitro data (Fig. 5), the

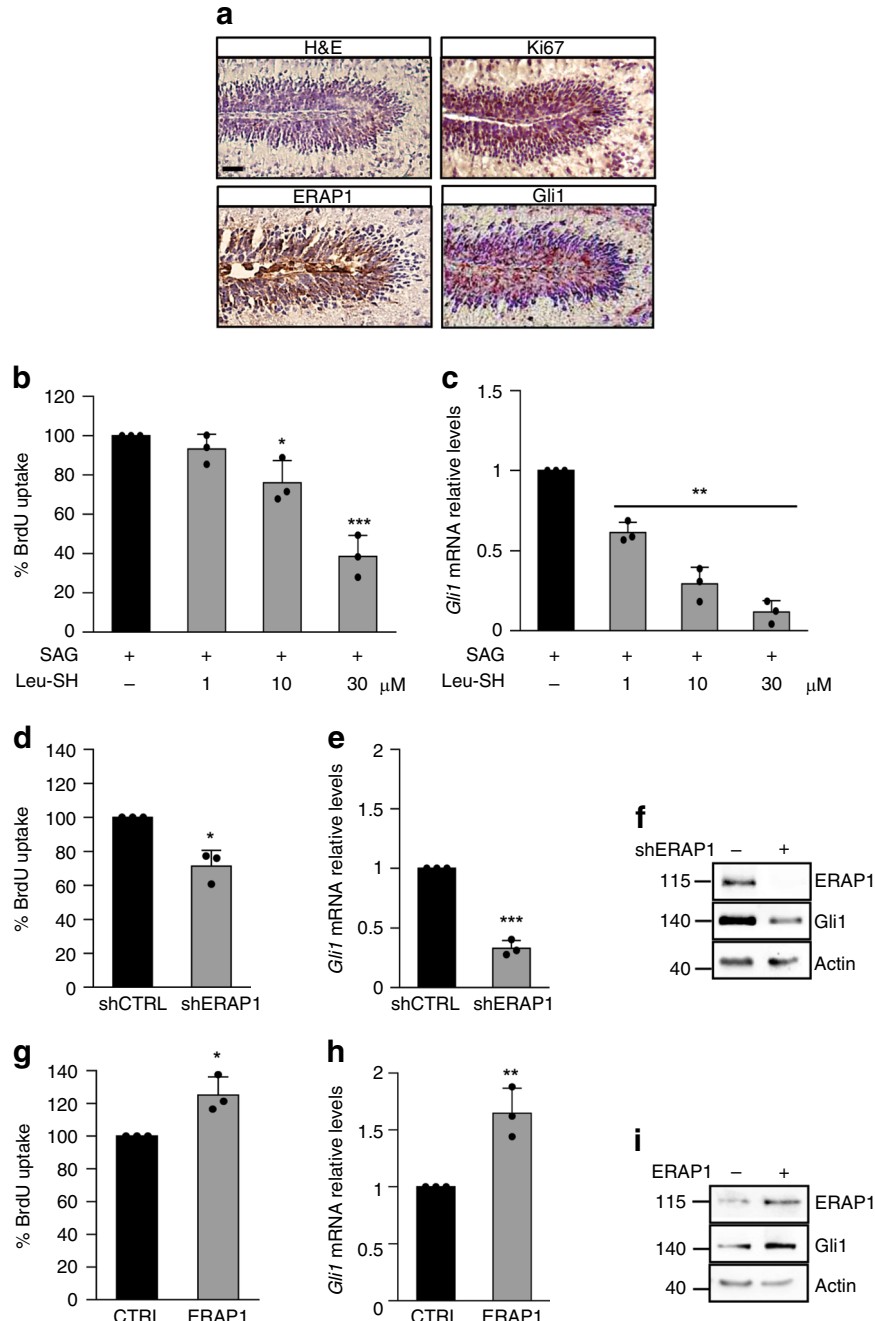

**Fig. 4** ERAP1 impairs Hh-dependent growth of cerebellar granule cell progenitors. **a** H&E and immunohistochemical staining of Ki67, ERAP1 and Gli1 in the outer EGL during mouse cerebellum development. Magnification ×40. Scale bars represent 50 μm. **b**, **c** GCPs were isolated from 4-day-old mice and treated with either SAG alone or in combination with increasing doses of Leu-SH for 24 h. BrdU uptake (**b**) and mRNA levels of *Gli1* (**c**) are shown. **d**, **i** GCPs isolated from 4-day-old mice were infected with lentiviral particles encoding for shERAP1 (**d–f**) or ERAP1 (**g–i**) and the corresponding controls, respectively. The percentage of BrdU uptake (**d**, **g**), mRNA (**e**, **h**), and protein levels (**f**, **i**) of Gli1 are shown. Results in **c**, **e**, **h** were normalized to endogenous *GAPDH* and *HPRT* controls and expressed as described in Fig. 1 legend, and represent the mean of three independent experiments. In f and i, actin was used as loading control. Mean ± S.D. *P < 0.05; **P < 0.01 determined with two-sided Student's *t*-test

expression of endogenous Hh target genes was reduced at both mRNA and protein levels in Leu-SH-treated tumors compared to controls, whereas protein levels of NeuN, cleaved Caspase-3 and βTrCP were increased (Fig. 6e, f). An orthotopic allograft animal model in which primary MB cells isolated from Math1-cre/Ptc^C/C mice tumors were implanted into the cerebellum of NSG mice confirmed these results. The animals treated with Leu-SH showed a significant reduction of tumor masses as compared to control (Fig. 6g, h). Accordingly, in a further

allograft model all mice engrafted on the flanks with MB cells infected with control lentiviruses developed progressively enlarging tumors, whereas no palpable tumor masses were developed in most mice engrafted with MB cells infected with shERAP1 lentiviruses (Fig. 6i, j). Consistent with above data, these tumor masses showed a reduced cellularity with few MB cells dispersed in a large amount of Masson's staining-mediated blue-labeled connective tissue and reduced Hh-pathway signature, as compared to control tumors (Fig. 6k, l). A more

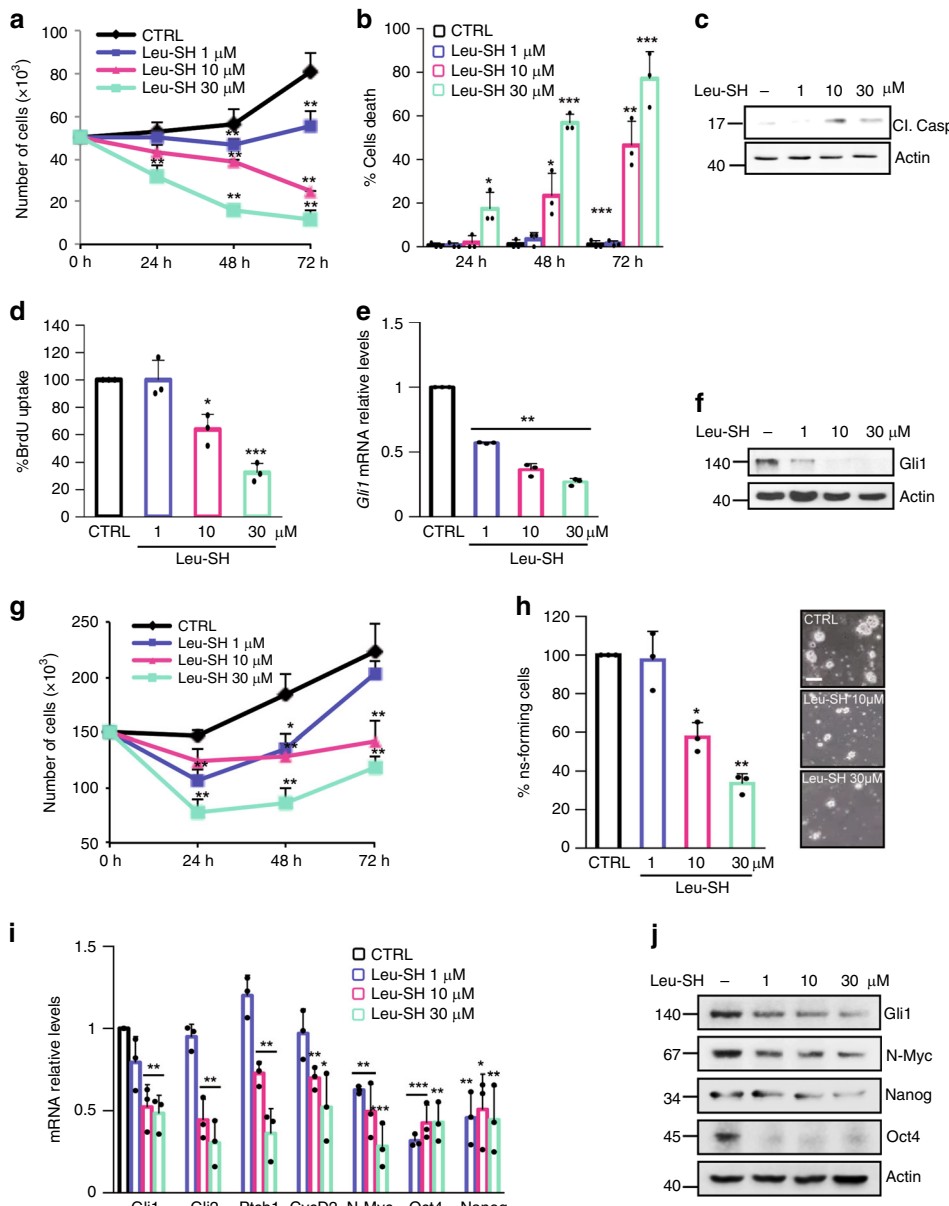

**Fig. 5** ERAP1 impinges Hh-dependent tumor cell growth in vitro. **a–f** Primary cell cultures from Math1-cre/Ptc$^{C/C}$ mice MBs were treated with different amounts of Leu-SH. **a, b** Cells were counted with trypan blue at the indicated time points to evaluate the growth rate of viable cells (**a**) and the percentage of cell death (**b**). **c** Cleaved Caspase-3 protein levels in cells treated with Leu-SH at the indicated concentration for 24 h. **d–f** Percentage of BrdU uptake (**d**) and *Gli1* mRNA (**e**), and protein (**f**) expression in MB cells treated with Leu-SH at the indicated concentrations for 24 h. **g** MB Stem-Like Cells (MB-SLCs) from Math1-cre/Ptc$^{C/C}$ mice were treated with Leu-SH as in (a) and counted with trypan blue at the indicated time points. **h** MB-SLCs were dissociated and treated with the indicated concentrations of Leu-SH or DTT as control. After 7 days of treatment, the number of secondary neurospheres derived from a known number of single cells was evaluated. The self-renewal MB-SLCs capability is expressed as percentage of neurosphere-forming cells (right). Representative bright field images of tumor neurospheres after Leu-SH treatment are shown (left). Scale bar 100 μM. **i, j** mRNA and protein expression levels of Hh target genes of MB-SLCs treated with the indicated concentrations of Leu-SH for 24 h. Actin was used as loading control. Results in **e, i** were normalized to endogenous *GAPDH* and *HPRT* controls and expressed as described in Fig. 1 legend. All data are representative of three independent experiments. Mean ± S.D. *P < 0.05; **P < 0.01 calculated using two-tailed Student's *t*-test

robust effect of the ERAP1 function on the MB in vivo cell growth was observed in an orthotopic allograft model where spontaneous primary MB cells from Math1-cre/Ptc$^{C/C}$ mice genetically silenced for ERAP1 were implanted into the cerebellum of NSG mice. As shown in Fig. 6m, only the cells infected with the lentiviral particles encoding for a non-targeting sequence gave rise to detectable tumor masses. These findings demonstrate that inhibition of ERAP1 interferes with Hh-dependent MB growth processes in vivo.

To investigate the role of ERAP1 during MB development, tumor-prone GCPs from the cerebellum of five-day postnatal (P5) Math1-cre/Ptc$^{C/C}$ mice were infected with lentiviruses expressing ERAP1 and then injected into the flank of NSG recipient animals. Compared to controls, mice engrafted with GCPs overexpressing ERAP1 showed an increased tumor growth rate, tumor volume (at the end point), and expression of *Gli1* and *CyclinD2* (Fig. 6n–p), accordingly with the role of ERAP1 in promoting Hh signaling.

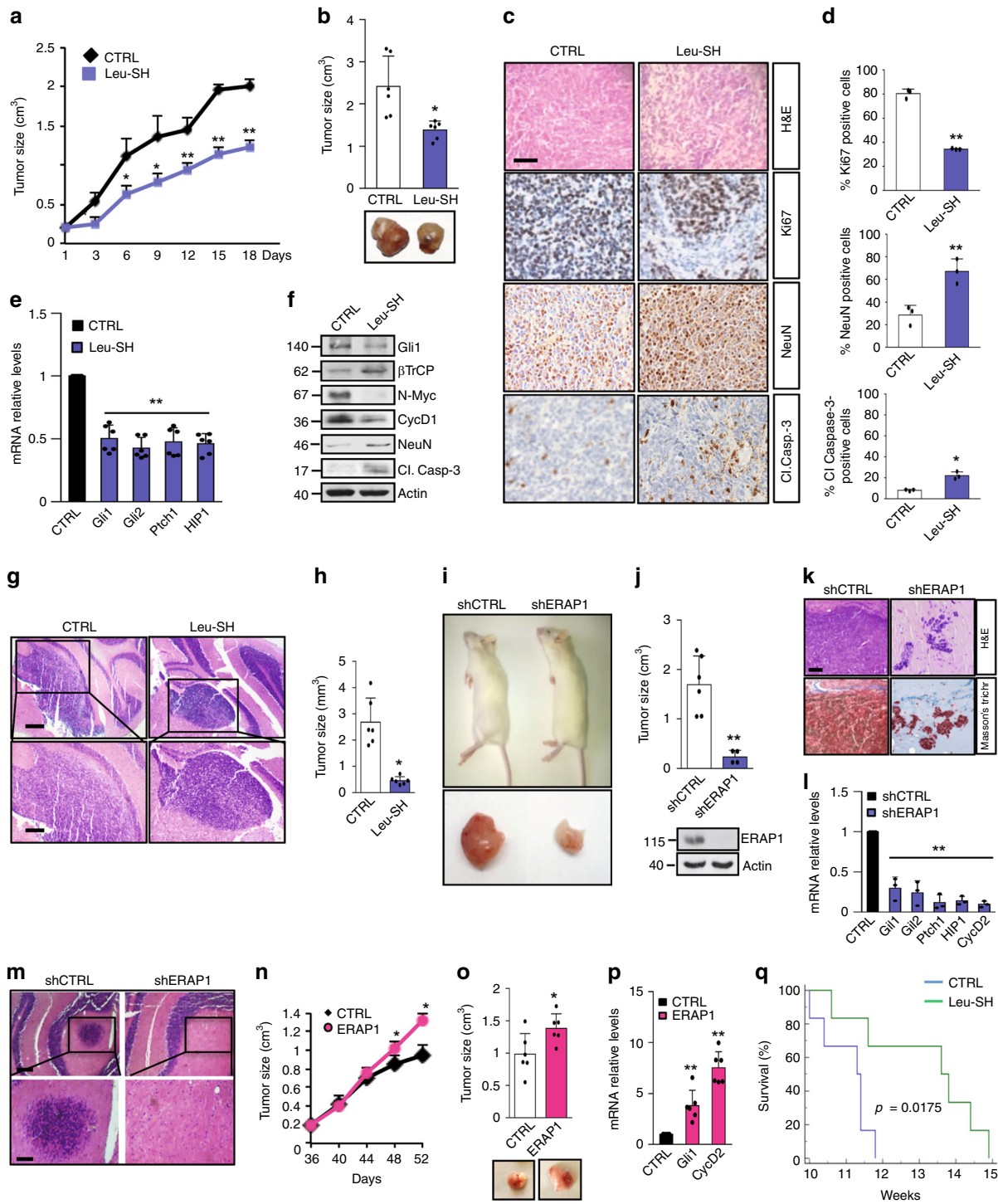

Moreover, to verify the effect of ERAP1 inhibition in a natural tumor niche for MB growth, symptomatic Gfap-cre/Ptc$^{fl/fl}$ mice were treated with Leu-SH or vehicle for two consecutive days. Treatment with Leu-SH resulted in reduced levels of the Hh pathway target genes and increased levels of βTrCP (Supplementary Figs. 6a, b). Interestingly, we found that i.p. treatment with Leu-SH significantly improved survival in the Math1-cre/Ptc$^{C/C}$ (Fig. 6q), demonstrating the potential therapeutic benefit of ERAP1 inhibition in the development of tumors in situ.

Together, these data identify a molecular mechanism in the regulation of Hh signaling and unveil the relevance of ERAP1 in the control of Hh-dependent tumor growth.

**ERAP1 inhibition impairs human SHH-MB growth.** Next, we investigated the effect of the inhibition of ERAP1 in human SHH-MB models. Similarly to murine tumor models, either pharmacological or genetic inhibition of ERAP1 impairs in vitro proliferation of the human Hh-dependent MB cell line Daoy (Supplementary Fig. 7a, b, d, e), leading to a decreased Hh-pathway activity (Supplementary Fig. 7c, f, g). To further evaluate the effect of ERAP1 activity in vivo, we performed a xenograft in NSG mice grafted on the flank with Daoy cells infected with lentiviral particles encoding for either shERAP1 or shCTRL. As expected, inhibition of ERAP1 significantly reduced tumor growth compared to controls, resulting in a decrease in

**Fig. 6** ERAP1 inhibition impairs Hh-dependent tumor growth in vivo. **a–f** NSG mice were grafted with spontaneous primary MB from Math1-cre/Ptc$^{C/C}$ mice. Tumor masses (150 mm$^3$) were intratumorally injected with Leu-SH. **a** Tumor growth was monitored. **b** Representative flank allograft average volumes (lower panel) and quantification of tumor explants (upper panel). **c**, **d** Ki67, NeuN, and cleaved Caspase-3 (Cl.Casp-3) immunohistochemical stainings of allograft tumor samples. **d** Quantification of immunohistochemical stainings shown in **c**. Scale bar 100 μm. **e** mRNA and **f** protein expression levels of Hh targets from tumors assayed in **b**. **g** Representative H&E images (low and high magnification) of a murine MB cell-derived orthotopic tumor in NSG mice after i.p. injection of Leu-SH. Scale bars, 500 μm and 200 μm (upper and lower panels, respectively). **h** Representative average volume of orthotopic tumor. **i–j** NSG mice were grafted with spontaneous primary MB from Math1-cre/Ptc$^{C/C}$ mice genetically silenced for ERAP1 expression. **i** Representative images of mice and the explanted tumor masses. **j** Quantification of the flank allograft average tumor volume. ERAP1 protein expression is shown below. In **f**, **j**, actin was used as loading control. **k** H&E and representative Masson's trichrome staining of tumors. Scale bar 100 μm. **l** mRNA levels of the indicated Hh target genes. **m** Representative H&E images (low and high magnification) of a murine MB cell-derived orthotopic tumor genetically interfered for ERAP1 before the injection in NSG mice cerebella. Scale bars, 500 and 200 μm (upper and lower panels, respectively). **n–p** ERAP1 accelerates Hh-MB formation. **n** Tumor volume of mice subcutaneously transplanted with GCPs from tumor-prone Math1-cre/Ptc$^{C/C}$ animals overexpressing ERAP1. **o** Representative flank allograft average volumes (lower panel) and quantification of the explanted tumor masses (upper panel). **p** mRNA expression of Hh target genes from the tumor masses assayed in **o**. **q** Survival curves of Math-cre/Ptc$^{C/C}$ mice treated with Leu-SH or vehicle. Results in **e**, **l**, **p** were normalized to endogenous *GAPDH* and *HPRT* controls and expressed as in Fig. 1. All data represent the mean of three independent experiments. Mean ± S.D. of tumor (n = 6) for each treatment. *P < 0.05, **P < 0.01, *** P < 0.001 calculated by two-sided Student's t-test

endogenous Hh target genes at both mRNA and protein levels (Supplementary Figs. 7h–k).

Finally, we found that the pharmacological inhibition of ERAP1 affected SHH-MB Patient-Derived Xenograft (PDX)[63] cell proliferation and increased tumor cell death in a dose- and time-dependent manner leading to apoptosis, as indicated by increased cleaved Caspase-3 positive cells (Fig. 7a–e). These results were confirmed by evaluating the cellular confluence over time of the SHH-MB PDX cells treated with increased amounts of Leu-SH (Supplementary Figs. 8a–c). As expected, the genetic depletion of ERAP1 in SHH-MB PDX cells resulted in reduced cell proliferation, increased cell death and reduced levels of Gli1 protein (Supplementary Figs. 8d–g).

In agreement with the in vitro data, Leu-SH treatment reduced the growth of MB SHH-PDX cells injected into the flank of NSG mice (Fig. 7f, g). As in the allograft model performed with spontaneous murine tumor (Fig. 6a–f), tumor masses treated with Leu-SH showed a reduced cellularity associated with a decrease in Ki67-positive tumor cells and an increase in NeuN and cleaved Caspase-3 positive tumor cells (Fig. 7h, i). Moreover, mRNA and protein expression levels of the endogenous Hh target genes were reduced, unlike the βTrCP protein levels (Fig. 7j, k). Overall, these data confirm the role of ERAP1 in the regulation of Hh-dependent tumor growth.

## Discussion

Hh signaling is an evolutionarily conserved pathway regulating cell fate and specification. Its tight regulation is essential for proper development and adult tissue preservation. Constitutive activation of the Hh pathway has been associated with a multitude of cancer types, including brain tumors. Hh signaling is also known to regulate stemness and drive tumor initiation and progression[64]. Due to its crucial role in tumorigenesis, Hh signaling has emerged as an attractive druggable target and a number of pathway-specific inhibitors are moving into the clinic[65,66]. However, therapeutic strategies aiming at blocking Hh signaling activation are complicated by the development of resistance and side effects, thus prompting alternative treatment approaches.

The combination of multiple drugs targeting different Hh signaling components and/or correlating Hh activating routes represents the most effective strategy for cancer treatment[67–72]. For this reason, the identification of molecular players controlling Hh activity is of clinical importance and represents a dramatic challenge in tumor biology.

In the present study, we identify and characterize an oncogenic property of ERAP1, an endoplasmic reticulum aminopeptidase. So

far, ERAP1 has been well-studied for its role in the antigen processing, a mechanism resulting in the production of high affinity peptides for the binding to MHC class I molecules. Like the other components of the antigen processing machinery, ERAP1 is induced in response to IFN-γ stimulation[22], a feature that makes it particular active in counteracting antiviral and anti-tumor immune responses. In this context, ERAP1 trims the N-terminal extension of precursor peptides to generate mature antigenic peptides[21]. Loss of ERAP1 function results in the generation of a new immunopeptidome, which stimulates anti-tumor immune responses determining the tumor regression of different mouse models[27,36]. Moreover, recent genome-wide studies have strongly associated ERAP1 polymorphisms with several autoimmune diseases, such as ankylosing spondylitis[29]. The polymorphic residues map to ERAP1's catalytic and regulatory sites and alter peptide specificity and processing activity, thus suggesting that the enzymatic activity of ERAP1 is important in the link with genetic disease[30]. However, the lack of commercial availability for highly specific chemical compounds for ERAP1 has constrained the progress in this area. Given the great interest in ERAP1, many pharmaceutical companies are investing in the development of ERAP1 inhibitors for potential therapeutic intervention.

Herein, we uncovered an unexpected role of ERAP1 in regulating Hh-dependent tumorigenesis, thus providing further evidence that inhibition of ERAP1 may be exploited for cancer treatment. We demonstrated that ERAP1 enhances Hh activity by sequestering USP47 and promoting ubiquitylation and degradation of βTrCP. This event results in an increase of Gli1 and Gli2 protein levels and a reduction of the Gli3R form, thus activating the Hh pathway and stimulating cell proliferation and tumorigenesis (Fig. 8). Conversely, inhibition of ERAP1 function stabilizes βTrCP, which in turn induces ubiquitylation of Gli factors leading to proteolysis of Gli1 and Gli2 and generation of Gli3R, thereby suppressing tumor cell growth.

Gli transcription factors are crucial effectors of the Hh pathway and their activity and expression are finely regulated by several mechanisms, mainly by the ubiquitin-proteasome system. Upon phosphorylation by PKA, GSK3β, and CK1, Gli factors are recognized by the SCF$^{βTrCP}$ ubiquitin ligase, which triggers the proteasome-dependent degradation of Gli1[9] and processing of Gli3 in its repressor form[14,41]. Gli2 is processed by both events, although the degradation one being predominant[8,10]. Hence, inhibition of the βTrCP-mediated degradation of Gli proteins is part of an Hh-induced activation signal by which Hh supports the function of Gli. Although βTrCP is fundamental for the regulation of Hh-mediated signals, the mechanisms controlling its function are still largely unexplored.

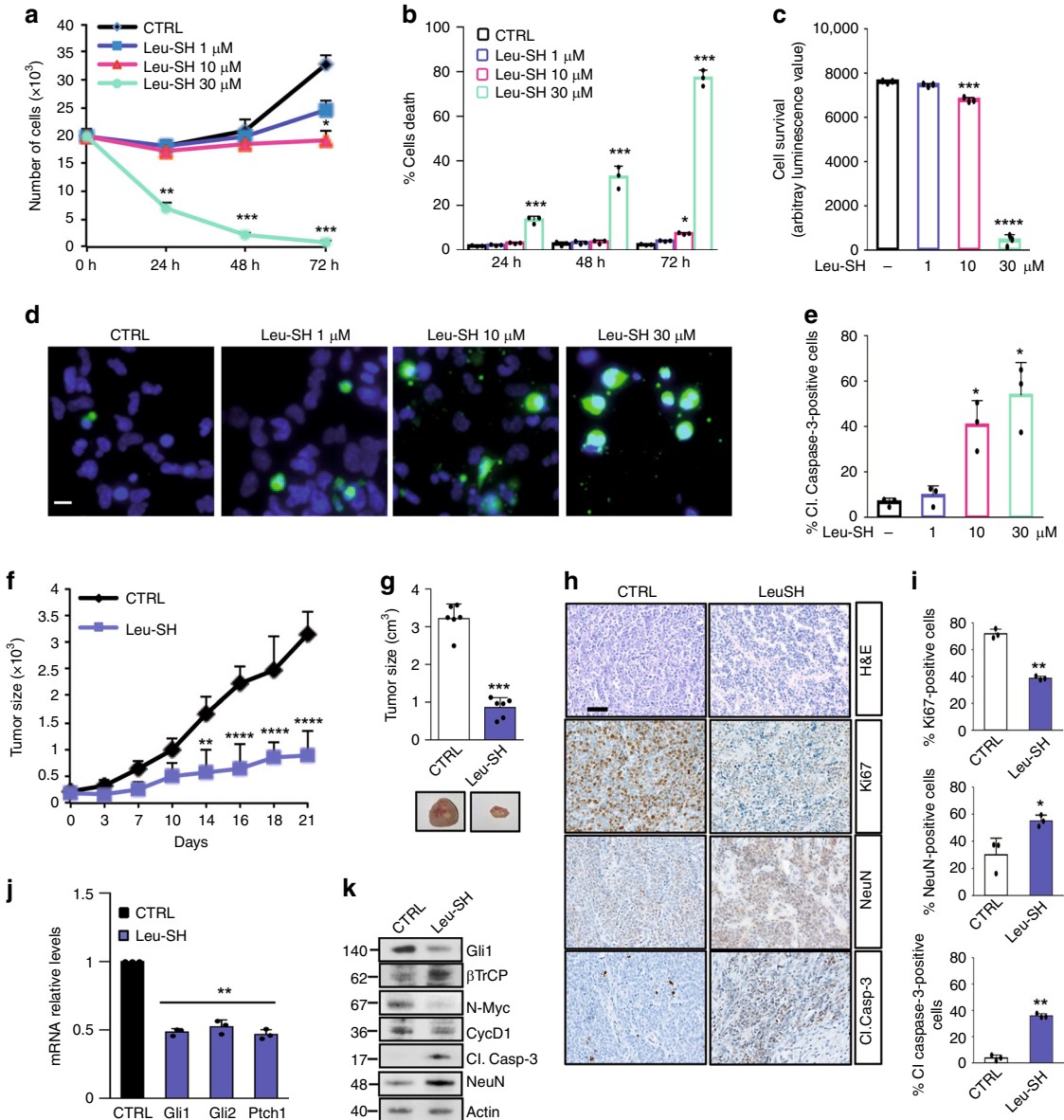

**Fig. 7 a–e** SHH-MB PDX tumor cells were treated with different amounts of Leu-SH. **a**, **b** Cells were counted with trypan blue at the indicated time points to evaluate the growth rate of viable cells (**a**) and the percentage of cell death (**b**). **c** Cell survival was also assessed by the Cell Titer Glo assay. **d** Immunofluorescent staining of cleaved Caspase 3 (green) of Leu-SH treated cells (48 h). Nuclei were counter stained with Hoechst (Blue). Scale bar 5 μm. **e** Quantification of cleaved Caspase-3 from immunofluorescence stainings in (**d**). (**f–k**) NSG mice ($n = 6$ for group) were grafted with SHH-MB PDX tumor cells. Tumor masses (150 mm$^3$) were intratumorally injected with Leu-SH (0.528 μM/Kg). **f** Tumor growth was monitored every three days. **g** Quantification of the explanted tumor masses (upper panel) and representative flank xenograft average volumes (lower panel). **h** H&E, Ki67, NeuN and cleaved Caspase-3 immunohistochemical staining of PDX tumor samples. **i** Quantification of Ki67, NeuN and cleaved Caspase-3 from immunohistochemical stainings in **h**. Scale bar 100 μm. **j**, **k** mRNA (**j**) and protein (**k**) expression levels of the indicated Hh target genes from the tumor masses assayed in **g**. Mean ± S.D. *$P < 0,05$; **$P < 0,01$; ***$P < 0,001$; ****$P < 0,0001$ determined using two-tailed Student's $t$-test

USP47 is a member of a class of enzymes named Ubiquitin-Specific Proteases (USPs), which are able to catalyze the removal of ubiquitin from substrates counteracting the activity of E3 ubiquitin ligases or protect them from self-ubiquitylation/degradation events.

Previous reports have demonstrated that USP47 is a key player in the regulation of cell viability and maintenance of genome integrity by promoting the stability of DNA polymerase β[52,73]. USP47 has been also described as a βTrCP interactor, although the biological outcome of this interaction is controversial[52,74]. Here, we found that USP47 strongly stabilizes βTrCP preventing its ubiquitin-dependent degradation. Interestingly, this process is

inhibited by ERAP1 that, through the binding to USP47, hampers the βTrCP/USP47 interaction and induces βTrCP proteolysis. Consequently, the Hh pathway is triggered by the transcription factors Gli, which promote cell proliferation. These results also provide convincing evidence that USP47 impairs Hh/Gli signaling and the Hh-driven tumorigenesis. Interestingly, no difference in ERAP1 and USP47 expression levels was observed in SHH MB as compared to other molecular subgroup or other brain tumor entities (Supplementary Fig. 9), thus suggesting that the activity, rather than the expression, of ERAP1 could be related to the SHH-MB. This adds more complexity due to the presence of modulators or not yet identified ERAP1 polymorphisms that

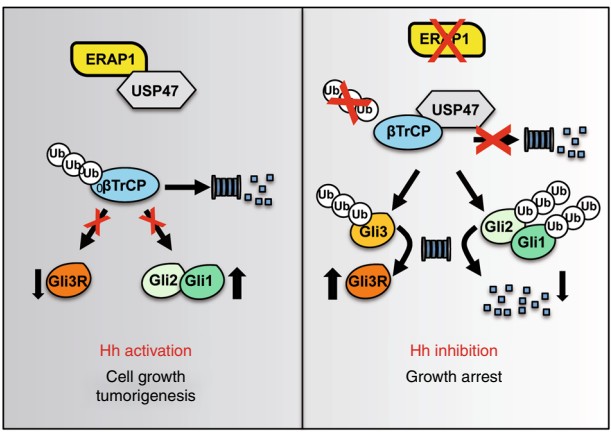

**Fig. 8** A representative model showing the role of ERAP1 in Hh-dependent tumorigenesis. ERAP1 promotes ubiquitylation and proteasomal degradation of βTrCP by sequestering USP47. This event leads to increase of Gli1 and Gli2 protein levels and decrease of Gli3R, thus triggering the Hh pathway and favoring cell growth and tumorigenesis. In the absence of ERAP1, USP47 binds and stabilizes βTrCP, which, in turn, promotes ubiquitylation and proteasomal degradation of Gli1 and Gli2, and ubiquitylation and proteolytic cleavage of Gli3 into the repressor form Gli3R. These events lead to the repression of the Hh pathway and inhibition of cell proliferation and tumor growth

could alter its conformational state and induce defects in its open-close transitions[75], thus modifying the activity of ERAP1 or its binding affinity to other proteins (i.e. to favor interaction with USP47 that leads to the degradation of βTrCP and stability of Gli proteins).

In conclusion, our study reveals an unexpected function of ERAP1 in cancer development suggesting that targeting ERAP1 could open innovative perspectives for effective therapeutic approaches in the treatment of Hh-dependent tumors.

## Methods

**Cell cultures, transfections, and lentiviral infections.** NIH3T3 cells, Shh-Light-II cells and MEFs from wild-type (WT) or Ptch$^{-/-}$, SuFu$^{-/-}$, USP47$^{+/+}$ and USP47$^{-/-}$ mice were cultured in Dulbecco's Modified Eagle's Medium (DMEM) plus 10% fetal bovine serum (FBS) or 10% bovine serum (BS) for NIH3T3 cells. Daoy cells were cultured in Eagle's minimum essential medium (MEM) plus 10% FBS. All media contained L-glutamine and antibiotics. HEK293T (CTR-3216$^{TM}$) and NIH3T3 (CRL-1658$^{TM}$) cells were obtained from ATCC. SuFu$^{-/-}$ MEFs were gift from Dr. R. Toftgård (Karolinska Institute), Ptch$^{-/-}$ MEFs were gift from Dr. M. P. Scott (Stanford, California, USA). For cerebellar GCPs culture from 4-days-old mice, cerebella were removed aseptically, cut into small pieces, and incubated at room temperature for 15 min in digestion buffer [Dulbecco's PBS (Invitrogen, Gaithersburg, MD) with 0.1% trypsin, 0.2% EDTA, and 10 μg/ml DNase]. Tissues were then triturated with fire-polished Pasteur pipettes to obtain a single-cell suspension. Cells were centrifuged and resuspended in Neurobasal medium supplemented with B27 (2%), penicillin–streptomycin (1%) and L-glutamine (1%) (Invitrogen) and plated at a density of $8 \times 10^5$ cells/cm$^2$. Primary MB cells were freshly isolated from Ptch$^{+/-}$ mice. Briefly, tumor was mechanically disrupted with fire-polished Pasteur pipettes in HBSS with 1% Pen/Strep and treated with DNase (10 μg/ml) for twenty minutes. Cells were centrifuged and resuspended in Neurobasal Media-A with B27 supplement minus vitamin A, penicillin–streptomycin (1%) and L-glutamine (1%). Stable Hh-dependent MB cells were cultured as neurospheres in DMEM/F12 media (2% B27 minus vitamin A; 3% Glucose 10×; 0.2% Insulin 10 mg/ml; 1% Pen/Strep; 0.01% Heparin 2 mg/ml; 0.06% N-Acetyl-L-Cysteine).

Mycoplasma contamination in cell cultures was routinely detected by using PCR detection kit (Applied Biological Materials, Richmond, BC, Canada).

Transient transfections were performed using DreamFect$^{TM}$Gold transfection reagent (Oz Biosciences SAS, Marseille, France), or Lipofectamine® with Plus$^{TM}$ Reagent (Thermo Fisher Scientific, Waltham, MA, USA) in accordance with the manufacturer's protocols.

Lentiviral particles were generated in HEK293T cells by combining packaging plasmids pCMV-dR8.74 and VSV-G/pMD2, with pLKO.1 plasmid (shCTRL SHC002; shERAP1 TRCN0000031119 (#1), TRCN0000031121 (#2), TRCN0000060542 (for Daoy and PDX), Sigma-Aldrich) or TWEEN-ERAP1 and

its empty vector, using TransIT-293 transfection reagent (MIRUS Bio LLC, Madison, WI, USA). NIH3T3, MEFs and Daoy cells were infected by spin inoculation method. Primary MB cells were infected with purified lentiviruses for 72 h.

For RNA interference, cells were transfected with scrambled or βTrCP shRNAs (Cat no: D-001810-10-05 and E-044048-00, respectively, Dharmacon, Inc., Lafayette, CO, USA) for 48 h with HiPerFect transfection reagent (Qiagen, Hilden, Germany) according to the manufacturer's instructions.

**Plasmids, Antibodies and other reagents.** pcDNA3.1-Flag-USP47, pcDNA3-Flag-Gli1, pcDNA3-Flag-Gli2, and pcDNA3-Flag-Gli3 were generated in our lab with standard cloning techniques and verified by sequencing. pCS2HA3hGli1 WT and pCS2HA3hGli1ΔC were kindly provided by A.E. Oro. pCMV6-XL5-ERAP1 (SC311137) was purchased from Origene (Rockville, MD, USA). shCTRL (SHC002) and shERAP1 (TRCN0000031119, TRCN0000031121, TRCN0000060542) in pLKO.1 plasmids were purchased from Sigma-Aldrich. ERAP1 transcript variant 2 was cloned in the lentiviral vector pRRL-CMV-PGK-GFP-WPRE (TWEEN) under the control of the CMV promoter.

Mouse anti-Gli1 (L42B10, 1:500), rabbit anti-βTrCP (D13F10, 1:1000), rabbit anti-cleaved Caspase-3 (Asp175 D3E9, 1:100 for IHC, 1:500 for WB) and rabbit anti-Phospho-PKA Substrate (RRXS*/T*, 100G7E, 1:1,000) were purchased from Cell Signaling (Beverly, MA, USA). Mouse anti-α Tubulin TU-02 (sc-8035, 1:1000), goat anti-Actin I-19 (sc-1616, 1:1,000), mouse anti-βTrCP C-6 (sc-390629, 2 μg), mouse anti-HA-probe F-7 HRP (sc-7392 HRP, 1:1,000), mouse anti-pCAF E-8 (sc-13124, 1:500), rabbit anti-Cyclin D1-20 (sc-717, 1:500), mouse anti-N-Myc B8.4.B (sc-53993, 1:500), rabbit anti-Gli1 H300 (sc-20687, 1:100) and HRP-conjugated secondary antibodies were purchased from Santa Cruz Biotechnology (Santa Cruz, CA, USA). Anti-Flag M2 HRP (A8592, 1:1000) and rabbit anti-Flag (F7425, 2 μg) were purchased from Sigma Aldrich (St Louis, MO, USA). Rabbit anti-USP47 (A301-048A, 1:1000, 2 μg) from Bethyl Laboratories (Montgomery, TX, USA). Mouse anti-SKP2 (323300, 1:500) were purchased from Invitrogen (Thermo Fisher Scientific, Waltham, MA, USA). Goat anti-Gli3 (AF3690, 1:1000) and goat anti-Gli2 (AF3635, 1:1000) were from R&D Systems (Minneapolis, MN, USA). Mouse anti-ERAP1 6H9 (1:1000) and mouse anti-ERAP1 4D2 (2 μg), kindly provided by P. van Endert, recognize denatured and native human ERAP1, respectively. Mouse anti-Itch (611199, 1:1000) antibody was purchased from BD Bioscience (Heidelberg, Germany). Rabbit anti-SPOP (16750-1-AP, 1:1000), Rabbit anti-USP22 (55110-1-AP, 1:2000) and Rabbit anti-USP24 (13126-1-AP, 1:1000) were purchased from Proteintech (Thermo Fisher Scientific, Waltham, MA, USA). Rabbit anti-Ki67 SP6 (MA5-14520, 1:100) was from Thermo Fisher Scientific (Waltham, MA, USA). Anti-rabbit Alexa Fluor 488 (A21206, 1:400 in BSA 3%) and anti-mouse Alexa Fluor 546 (A11003, 1:400 in BSA 5% and Goat Serum 3%) were purchased from Life Technologies (Foster City, CA, USA). Mouse anti-NeuN (clone A60, MAB377, 1:100 for IHC and 1:1000 for WB) was from Millipore (Merk, Darmstadt, Germany).

Where indicated, cells were treated with SAG (200 nM, Alexis Biochemicals Farmingdale, NY, USA) for 24 or 48 h, MG132 (50 μM; Calbiochem, Nottingham, UK) for 4 h, Cycloheximide (CHX 100 μg/ml, Sigma Aldrich), Dihydrochloride (H-89, Calbiochem, Nottingham, UK), L-Leucinethiol (Leu-SH, Sigma Aldrich) or Dithiothreitol (DTT, SERVA, Heidelberg, Germany) as indicated.

**Luciferase reporter assay.** The Hh-dependent luciferase assay was performed in Shh-Light II cells, stably expressing a Gli-responsive luciferase reporter and the pRL-TK Renilla (normalization control), treated for 48 h with SAG (200 nM) and for 24 h with Leu-SH and/or DTT as control at the indicated concentrations.

AP1/Jun- and WNT/β-Catenin-luciferase assays were carried out in MEFs WT transfected with MMP1-luciferase reporter and Jun- or Top Flash-luciferase reporter and β-Catenin, respectively, and pRL-TK Renilla. After 24 h from transfection, cells were treated with increasing amounts of Leu-SH and/or DTT.

Luciferase and Renilla activities were assayed with a dual-luciferase assay system according to the manufacturer's instructions (Biotium Inc., Hayward, CA, USA). Results were expressed as luciferase/Renilla ratios and represented the mean ± S.D. of three experiments, each performed in triplicate.

**Immunoblot analysis and immunoprecipitation.** Cells were lysed in a solution containing RIPA buffer (50 mM Tris-HCl at pH 7.6, 150 mM NaCl, 0.5% sodium deoxycholic, 5 mM EDTA, 0.1% SDS, 100 mM NaF, 2 mM NaPPi, 1% NP-40) supplemented with protease and phosphatase inhibitors. The lysates were centrifuged at 13,000 g for 30 min at 4 °C and the resulting supernatants were subjected to immunoblot analysis. Immunoprecipitation was performed using whole-cell extracts obtained by lysing cell pellets with Triton Buffer (50 mM Tris-HCl pH 7.5, 250 mM sodium chloride, 50 mM sodium fluoride, 1 mM EDTA pH 8, 0.1% Triton), supplemented with protease and phosphatase inhibitors. Cell lysates were immunoprecipitated overnight at 4 °C with rotation with specific primary antibodies or IgG used as a control (1–2 μg/ml, Santa Cruz Biotechnology) and then incubated with Protein G- or Protein A-agarose beads (Santa Cruz Biotechnology) for 1 h at 4 °C with rotation. The immunoprecipitates were then washed five times with the lysis buffer described above, resuspended in sample loading buffer, boiled for 5 min, resolved in SDS-PAGE and then subjected to immunoblot analysis.

Uncropped scans of the most important blots are reported in Supplementary Figs. 10 and 11.

**In vivo ubiquitylation assay**. MEFs were lysed with denaturing buffer (1% SDS, 50 mM Tris-HCl at pH 7.5, 0.5 mM EDTA, 1 mM DTT) to disrupt protein-protein interactions. Lysates were then diluted 10 times with NETN lysis buffer and subjected to immunoprecipitation with anti-βTrCP (Santa Cruz Biotechonology) overnight at 4 °C with rotation. The immunoprecipitated proteins were then washed five times with the NETN lysis buffer, resuspended in sample loading buffer, boiled for 5 min, resolved in SDS-PAGE and then subjected to immunoblot analysis. Polyubiquitylated forms were detected using mouse anti-HA from Santa Cruz Biotechnology.

**mRNA expression analysis**. Total RNA was extracted using TRIzol reagent (Thermo Fisher Scientific) and reverse-transcribed with SensiFASTcDNA Synthesis Kit (Bioline Reagents Limited, London, UK). Quantitative real time PCR (qPCR) analysis of *ERAP1*, *βTrCP*, *Gli1*, *Gli2*, *Ptch1*, *Hip1*, *CyclinD2*, *N-Myc*, *Oct4*, and *Nanog* mRNA expression was performed using the ViiA™ 7 Real-Time PCR System (Life Technologies). Standard qPCR thermal cycler parameters were used to amplify a reaction mixture containing cDNA template, SensiFAST™ Probe Lo-ROX mix (Bioline Reagents Limited) and Taqman Gene Expression Assays (Thermo Fisher Scientific). The average of three threshold cycles was used to calculate the amount of transcript in each sample amplified in triplicate (using SDS version 2.3 software). mRNA quantification was calculated as the ratio of the sample quantity to the calibrator quantity expressed in arbitrary units. Data were normalized with the endogenous controls (*GAPDH* and *HPRT*) and expressed as the fold change respect to the control sample value.

**BrdU incorporation and MB neurosphere-forming assay**. Cell proliferation was evaluated by BrdU detection (Roche, Welwyn Garden City, UK). Briefly, cells were pulsed 24 h with BrdU and then fixed and permeabilized with 4% paraformaldehyde and 0.2% Triton X-100, respectively. Nuclei were counterstained with Hoechst reagent and BrdU detection was performed according to the manufacturer's instructions. At least 500 nuclei were counted in triplicate, and the number of BrdU-positive nuclei was recorded. To determine the growth rate of viable cells, a trypan blue count was performed after a treatment period of 24, 48 and 72 h with Leu-SH at the indicated dose.

For the neurosphere-forming assay, cells were plated at clonal density (1–2 cells/mm$^2$) into 96-well plates and treated with Leu-SH at the indicated concentration.

**Cell viability assay in SHH-MB PDX model**. Patient-derived xenograft (PDXs)-ICN-MB 12 was generated from primary human SHH- MB sample diagnosed at the Children's Necker Hospital in Paris and transplanted into the subscapular fatpad of immunocompromised NOD/SCID mice[63]. Human sample for xenograft studies was obtained under written informed consent and ethical approved by the Internal Review Board of the Necker Sick Children's Hospital, Paris, France. Tumor cells from PDX model were purified and cultured[63]. For culture experiments, 75,000 tumor cells per well were plated in 96-well plates, pre-coated with poly-D-lysine (EMD Millipore, Billerica, MA) and Matrigel (BD Biosciences, San Jose, CA). Cells were grown in Neurobasal medium with B27 supplement, 2 mM glutamine, penicillin/streptomycin (all from Thermo Fisher Scientific), bovine serum albumin, and 0.45% D-glucose (both from Sigma Aldrich). The next day, cultured tumor cells were treated with different concentrations of Leu-SH plus 0.1 M DTT or water plus 0.1 M DTT for the control. For the Incucyte experiment, cells were treated with 0.3 μg/ml of propidium iodide (PI, Sigma Aldrich). Then, the plates were scanned for phase contrast and PI staining every 3 h during 72 h using the IncuCyte imager with a 4X objective (Essen BioScience). Proliferation was measured using quantitative kinetic processing metrics from time-lapse image acquisition and showed as percentage of culture confluence over time. For the PI staining, which allowed to fluorescently stain the nuclear DNA of cells that have lost plasma membrane integrity, the percentage of PI positive cells (corresponding to red object confluence) was divided by the phase object confluence percentage for each well, thus indicating the level of dead cells in each well. For the CellTiter-Glo® Luminescent Cell Viability Assay the cell viability was examined after 72 h of treatment according to the manufacturer's instructions (Promega Corporation, Madison, WI, USA). Results were expressed as luciferase fluorescence and represented the mean ± S.D. of three experiments, each performed in triplicate.

**Animal studies**. For allograft experiment, spontaneous MB from Math1-cre/Ptc$^{C/C}$ mice[57] was isolated, minced and pipetted to obtain a single-cell suspension. Equal amounts of cells ($2 \times 10^6$) were injected s.c. at the posterior flank of NSG (Charles River Laboratories, Lecco, Italy). Tumors were grown until a median size of ~150 mm$^3$. Animals were randomly divided in two groups ($n = 6$) and intratumorally injected every other day with (0.528 μmol/Kg) Leu-SH and/or 0.1 M DTT for 18 days. Primary cells of spontaneous MB from Math1-cre/Ptc$^{C/C}$ mice were infected for 72 h with purified lentiviral particles encoding short hairpin RNA targeting murine ERAP1 (shERAP1) or a control non-targeting sequence

(shCTRL). Equal amounts of cells ($2 \times 10^6$) were injected s.c. at the posterior flank of NSG mice. GCPs from the cerebella of postnatal (P) day P5-P7 Math1-cre/Ptc$^{C/C}$ mice were infected with purified lentiviral particles encoding ERAP1 or an empty GFP vector. Equal amounts of cells ($2 \times 10^6$) were injected s.c. at the posterior flank of NSG mice. Cells were resuspended in an equal volume of culture medium and Matrigel (BD Biosciences, Heidelberg, Germany) before the s.c. injection. After the injection, tumor growth was monitored and measured with caliper. Changes in tumor volume were evaluated with the formula (length × width) × 0.5 × (length + width).

For orthotopic allograft model, adult NSG mice were anesthetized by i.p. injection of ketamine (10 mg/kg) and xylazine (100 mg/kg). The posterior cranial region was shaved and placed in a stereotaxic head frame and primary cells of spontaneous MB from Math1-cre/Ptc$^{C/C}$ mice freshly isolated or infected for 72 h with purified lentiviral particles encoding murine shERAP1 or shCTRL were stereotaxically implanted into the cerebellum ($2 \times 10^5$/3 μl) according to the atlas of Franklin and Paxinos coordinates. After injection, at an infusion rate of 1 μl/min, the cannula was kept in place for 5 min and then the skin was closed using metallic clips. After 10 days following tumor implantation, the animals were randomly divided into two groups ($n = 6$) and treated i.p. every other day with Leu-SH 1 mg/Kg or vehicle only. After 25 days of treatment, animals were sacrificed and brains were fixed in 4% formaldehyde and paraffin embedded. Mice implanted with the tumor cells silenced for ERAP1 were sacrificed after six weeks. Tumor volume calculation was performed on serial 40 coronal sections of 2 μm after H&E staining every 40 μm of brain slice. A microscope (Axio Imager M1 microscope; Leica Microsystems GmbH, Wetzlar, Germany) equipped with a motorized stage and Image Pro Plus 6.2 software was used to evaluate tumor area of each slide and the tumor volume was calculated by the formula: tumor volume = sum of measured area for each slice × slice thickness × sampling frequency[59].

For Gfap-Cre/Ptc$^{fl/fl}$ mice injection, symptomatic Gfap-Cre/Ptc$^{fl/fl}$ mice were randomly divided into two groups ($n = 5$) and injected s.c under the scruff with (0.528 μmol/Kg) Leu-SH and/or 0.1 M DTT for two days. Tumor masses were analyzed by qRT-PCR and immunoblotting. For survival analysis, P21 Math1-cre/Ptc$^{C/C}$ mice were randomized to receive either Leu-SH (1 mg/Kg) or vehicle every other day by i.p. injection. Statistical analysis was performed by MedCalc software.

For xenograft experiment, Daoy cells were infected with lentiviral particles encoding either short hairpin RNA targeting human ERAP1 (shERAP1) or a control non-targeting sequence (shCTRL). Equal amounts ($2 \times 10^6$) of cells were injected s.c. at the posterior flank of NSG mice.

Patient-derived xenograft (PDXs)-ICN-MB 12 was isolated, minced and purified. Cells ($3 \times 10^6$) were injected s.c. at the posterior flank of NSG mice. Tumors were grown until a median size of ~150 mm$^3$. Animals were randomly divided in two groups ($n = 6$) and intratumorally injected with (0.528 μmol/Kg) Leu-SH and/or 0.1 M DTT for 18 days. Cells were resuspended in an equal volume of culture medium and Matrigel before the s.c. injection. Tumor growth was monitored and measured with caliper. Changes in tumor volume were evaluated with the formula (length x width) x 0.5 × (length + width).

All animal protocols were approved by local ethic authorities (Ministry of Health) and conducted in accordance with Italian Governing Law (D.lgs 26/2014).

**Immunohistochemistry**. Formaldehyde-fixed paraffin-embedded (FFPE) tissues and frozen OCT-embedded tissues were cut into 4μm sections for Gli1, Ki67, cleaved Caspase-3, NeuN, and ERAP1 immunohistochemical staining. FFPE slides were deparaffinized and subjected to heat-induced antigen retrieval at low or high pH buffer, whereas frozen sections were fixed in 4% paraformaldehyde. Slides were blocked for 30 min with 5% PBS/BSA. FFPE slides were incubated with monoclonal antibodies against Gli1, Ki67, cleaved Caspase-3, NeuN, whereas cryostat sections were incubated with monoclonal antibody ERAP1 (4D2, 50 mg/ml overnight 4 °C). This step was followed by incubation for 20 min with secondary antibodies coupled with peroxidase (Dako). Bound peroxidase was detected with diaminobenzidine (DAB) solution and EnVision FLEX Substrate buffer containing peroxide (Dako). Cell quantification was performed on collected sections using the imaging software NIS-Elements BR 4.00.05 (Nikon Instruments Europe B.V., Italy). Images were captured by HistoFAXS software (TissueGnostics GmbH, Vienna, Austria) at 20x magnification. Tumor regions were analyzed with HistoQuest software (TissueGnostics) for automatic color separation and quantification. Expression levels were evaluated as stained area per mm$^2$.

**Immunofluorescence**. After 24 h of transfection, cells were fixed for 15 min in 4% paraformaldehyde, treated with glycine 1 M for 15 min to saturate the residual site of paraformaldehyde, permeabilized for 8 min in 0.2% Triton X-100 and blocked with 3% BSA for 30 min. Cells were then labeled with primary antibody for 1 h, followed by staining with secondary antibodies specific for rabbit or mouse (Alexa Fluor 488, A21206, and Alexa Fluor 546, A11030, respectively, Life Technologies). Single plane confocal images were acquired using an inverted Olympus iX73 microscope equipped with an X-light Nipkow spinning-disk head (Crest Optics, Rome, Italy) and Lumencor Spectra X Led illumination. Images were collected using a CoolSNAP MYO CCD camera (Photometrics, Tucson, AZ, USA) and MetaMorph Software (Molecular Device, Sunnyvale, CA, USA) with a x60 oil objective.

**Statistical analysis**. Statistical analysis was performed using the StatView 4.1 software (Abacus Concepts, Berkeley, CA, USA). For all experiments, $P$ values were determined using two-tailed Student's $t$-test and statistical significance was set at $P < 0.05$. Results are expressed as mean ± S.D. from an appropriate number of experiments (at least three biological replicas). For IncuCyte experiments, statistical significance was determined with GraphPad Prism software (version 6.0, La Jolla, CA, USA). Data were analyzed with the Two-way ANOVA test and given as mean ± SD. For survival analysis statistical significance was calculated with Logrank-test performed by MedCalc software.

## Data availability

All data in this study are available within the Article and Supplementary Information or from the corresponding author on reasonable request.

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

## Acknowledgements

We thank M.P. Scott for the gift of Ptch$^{-/-}$ MEFs, R.Toftgård for SuFu$^{-/-}$ MEFs, Peter van Endert for ERAP1 antibodies, Claire Lovo, from the PICT-IBiSA Orsay Imaging facility of Institut Curie, for IncuCyte imaging assistance, and C. Alberti and E. Belloir for in vivo experiments at the Institut Curie mouse facilities, C. Felici, I. Basili and A. Scipioni for their experimental assistance. This work was supported by Associazione Italiana Ricerca Cancro (AIRC) Grants #IG14723 and #IG20801 to L.D.Mar, #IG18495 to D.F., #IG17575 to G.C. and #IG17734 to G.G., Progetti di Ricerca di Università Sapienza di Roma, Italian Ministry of Health Grant PRIN 2012C5YJSK_002 and PRIN 2017BF3PXZ_003 to L.D.Mar. and #PE-2011-02351866 to D.F., Italian Ministry of Education, Universities and Research - Dipartimento di Eccellenza - L. 232/2016, Pasteur Institute/Cenci Bolognetti Foundation, Istituto Italiano di Tecnologia (IIT). LLS was supported by PhD Degree Program in Biotechnology in Clinical Medicine, University of Rome La Sapienza.

## Author contributions

L.D.Mar. conceived and coordinated the project, designed experiments, analyzed the data, and wrote the paper. F.Bu., P.I. and L.D.Mar. conceived, performed experiments, and analyzed the data. F.Bu., P.I., F.Be. and M.C., performed most of the experiments. P.R., F.Bu. O.M., and L.L.S. generated lentivirus and analyzed data. P.I., M.M., L.L.S., L.D. Mag and M.T. performed the animal experiments, IHC and analysis. J.T., S.Pu. and O.A. provided and performed experiments on SHH-PDX. A.P., L.B., E.D.S., G.C., D.G., D.B., C.C., G.M., F.L., S.Pa., O.A., G.G., A.G. and D.F. discussed the results, and provided critical reagents and comments. F.Bu., D.F. and L.D.Mar. wrote the paper. All authors critically revised and edited the paper.
