## [Peer Review File · Nature Communications]

Reviewers' comments:

Reviewer #1, Expertise: Hh signalling (Remarks to the Author):

In this manuscript the authors examine the role of endoplasmic reticulum aminopeptidase 1 (ERAP1) in the regulation of the Hedgehog (Hh) pathway and as a potential therapeutic target for Hh driven tumors. The authors demonstrate that knockdown or inhibition of ERAP1 attenuates the activation of Hh target genes, while overexpression of ERAP1 leads to elevated levels of Gli1. These changes correlate with increased levels of betaTrCP and Gli3 repressor (inhibition of ERAP1) and decreased levels of betaTrCP and Gli3 repressor (overexpression of ERAP1).

The loss of betaTrCP is due to its ubiquitination and degradation by the proteasome. A potential explanation for this turnover is their finding that ERAP1 binds the deubiquitinase USP47 and impairs its interaction with betaTrCP.

The effects of ERAP1 inhibition and ERAP1 overexpression were also observed in the developing cerebellum and Ptc mutant medulloblastoma tumors where the Hh pathway drives cell proliferation.

These are interesting results.

My main concern is with the underlying molecular mechanism. If ERAP1 regulation of the Hh pathway were going through betaTrCP, one would not expect it to have much of an effect in Ptc mutant MEFs where Smo is active and the betaTrCP degron should not be phosphorylated. Yet, inhibition of ERAP1 has a very strong effect. I think it is important to demonstrate that the effects of ERAP1 depend upon the presence of the betaTrCP degron in the Gli proteins.

The authors should also examine whether ERAP1 has any effects on SPOP as this E-3 ligase complex is also known to regulate the Gli proteins.

Reviewer #2, Expertise: Ubiquitin(Remarks to the Author):

Bufalieri and colleagues described a novel role for ERAP1 as a positive regulator of the Hedgehog signaling and as a promising target for the treatment of Hh-dependent tumors. The authors explored the molecular basis by which ERAP1 controls the Hh-pathway, focusing on the USP47- β TrCP circuitry. They demonstrated that ERAP1 physically interacts with the β TrCP-associated deubiquitylase enzyme USP47, thus, promoting β TrCP degradation and, consequently, Gli factors stabilization. Based on these results, the authors evaluated the possibility to target ERAP1 to impair Hh-driven tumor growth. The inhibition of ERAP1 shows to have a significant impact on tumor growth both in vitro and in vivo.

The central findings of this work are provocative and of solid, helping our understanding of the regulation of the Hh pathway and providing insights for the clinically implications of targeting Hh pathway. In particular, the study establishes an interesting and novel role for ERAP1 in the control of tumorigenesis, independently from its function in the immune system. Moreover, this study added valuable information about the involvement of upstream signals that control β TrCP-mediated degradation of Gli1, an important nodal point in Hh signaling regulation. Overall, the experiments are of high quality and the paper is clearly written. This reviewer appreciates the novelty of this work and believes it would warrant publication after addressing the points raised below:

- Figure 2a-c: the authors show the effects of either ERAP1 overexpression or its pharmacological inhibition on the protein levels of Gli factors. To make the results more homogeneous, the authors

should show also in Figure 2f the protein levels of Gli2 and Gli3.

- The authors shown that the modulation of ERAP1 affects protein levels and stability of β TrCP. In light of these data, it would be important analyze the effect of ERAP1 inhibition on other β TrCP substrates.

- Figure 3: the authors explain the molecular mechanism by which ERAP1 affects β TrCP stability (i.e., through the interaction with USP47). In Figure 6, the effect of ERAP on the Hh-dependent tumor growth was established by inhibiting ERAP1 both pharmacologically and genetically. Is the interaction between USP47 and β TrCP dependent on ERAP's activity? The authors should address this point by verifying the formation of β TrCP/USP47 complex following pharmacological inhibition of ERAP1 by Leu-SH treatment. Similarly to what shown in Figure 3l, one would expect an increase of the β TrCP/USP47 interaction after inhibition of ERAP1 activity.

- The effect of USP47 knockdown on β TrCP protein levels should be investigated in USP47^{-/-} MEFs after reintroduction of USP47. Further, Gli protein levels should be added in Figure 3f.

- The authors have provided several evidences of the effect of ERAP inhibition on Hh-dependent tumor growth, both in vitro and in vivo using medulloblastoma cells derived from Ptc^{+/+} mice. Does any medulloblastoma cell line behave similarly?

- Since the strong results obtained in SHH-MB Patient-Derived Xenograft culture, the authors might want to verify the effect of ERAP1 modulation on medulloblastoma PDX growth also in vivo.

Reviewer #3, Expertise: Medulloblastoma (Remarks to the Author):

Bufalieri et al present a manuscript where ERAP1 is shown to promote HH-dependent brain tumorigenesis by controlling USP47-mediated degradation of U3 ligase, Beta-TrCP. They show that ERAP1 physically binds to USP47 and thereby promoting Beta-TrCP degradation that normally control Gli protein stability. They further suggest using ERAP1 inhibitors to suppress SHH tumor formation and present this both in vitro and in vivo. The manuscript is well-written and experiments are solid. I still have a few concerns about the conclusions made from the obtained results.

Major comments:

1. What was the initial reason to study ERAP1 in HH tumors in particular in the first place? This is not clearly motivated in the introduction section.
2. Although the shERAP1 construct used in the manuscript shows great downregulation of ERAP1 as compared to shControl, it would still be good to test at least one additional shERAP1 construct to confirm similar downregulation and further rule out off target effects in Fig 1.
3. In Fig. 1d-h it is not clear how the expression levels are normalized and to what.
4. Experiments and results in Figs 2-4 are well performed and mechanistically relevant. However, it is not clear how specific Leu-SH is in targeting ERAP1 as compared to other aminopeptidases. A blot for other APs would have been explanatory. For example, only the highest concentrations 10-30uM of Leu-SH are regulating ERAP1 itself. Protein levels are not reduced in 1uM concentrations perhaps suggesting that the drug is not that specific.
5. In Fig 3, USP47 is picked and suggested as a DUB that would regulate Beta-TrCP. However other potential DUBs, e.g. USP24 (Wang et al. Nat Comm., 2018) that might regulate Beta-TrCP were not investigated, why?
6. In Fig 5b authors describe how cell death is increased but not what type of cell death (apoptosis, senescence, necrosis, cell cycle arrest, etc.).
7. Authors show significant reduction in cell growth in Fig 5j as a function of confluency over time. However, image-based growth assessment has several disadvantages such as local differences in density etc. Cell number counting, impedance or cell-metabolism assays might be more appropriate in this case.
8. In Fig 5k authors demonstrate late apoptotic events by PI staining. Annexin 5 co-staining and FACS analysis might supplement the data in the better way to detect early apoptotic events at lower concentrations.

9. In Fig 6c authors indicate reduced proliferation and increased cellularity of the tumors. This is a very promising finding, however does not indicate by which means the effect was achieved. Apoptosis marker analysis e.g. Cl. Cas 3 staining in combination with staining for neuronal differentiation markers such as SYP or NeuN shall be investigated.

10. The authors use in vivo allografts from a mouse SHH MB model harbouring Ptch1 mutation. In vivo experiments with a flank tumor that does not include a natural tumor niche for MB growth. A mouse model that spontaneously develops SHH MB or orthotopic transplants of the tumor would have been more relevant for the pharmacological studies. In addition, a survival curve of the treated mice would be beneficial for the significance of their data.

11. In order to increase the clinical relevance of the story, the manuscript would benefit from checking levels of USP47 and ERAP1 and see if they are differentially expressed in SHH MB patients as compared to other molecular subgroups of MB tumors or other brain tumor entities. Such data is readily and publically available.

Minor comments:

1. In Supplementary Fig. 1b it seems like there is a trend in increasing luc activity. I would suggest further increase in Leu-SH to confirm the hypothesis that the ERAP1 inhibition does not affect the beta-Catenin pathway.

2. From a clinical perspective, ERAP inhibition will probably lead to immune escape suggesting it would be a bad target for certain cancers and for tumor recurrence. Further, if aminopeptidase inhibitors like Leu-SH at all would be potential drug candidates in children is not clear. Any of these potential limitations should be included in the discussion.

3. Figure 4 and Supplementary Figure 4 are exactly the same.

4. In Fig5 the effect of ERAP1 inhibition on Oct4 are remarkable and do not follow pharmacological patten of other targets that were tested. Further follow-up on possible of target regulation or commenting on the explanation of this would be appropriate.

5. In the last section the heading is wrong: "ERAP1 inhibits Hh-dependent tumor growth in vivo".

Responses to Reviewers' comments:

Reviewer #1, Expertise: Hh signalling (Remarks to the Author):

In this manuscript the authors examine the role of endoplasmic reticulum aminopeptidase 1 (ERAP1) in the regulation of the Hedgehog (Hh) pathway and as a potential therapeutic target for Hh driven tumors. The authors demonstrate that knockdown or inhibition of ERAP1 attenuates the activation of Hh target genes, while overexpression of ERAP1 leads to elevated levels of Gli1. These changes correlate with increased levels of betaTrCP and Gli3 repressor (inhibition of ERAP1) and decreased levels of betaTrCP and Gli3 repressor (overexpression of ERAP1). The loss of betaTrCP is due to its ubiquitination and degradation by the proteasome. A potential explanation for this turnover is their finding that ERAP1 binds the deubiquitinase USP47 and impairs its interaction with betaTrCP. The effects of ERAP1 inhibition and ERAP1 overexpression were also observed in the developing cerebellum and Ptc mutant medulloblastoma tumors where the Hh pathway drives cell proliferation. These are interesting results.

We thank the Reviewer for the positive comment on our study.

1. My main concern is with the underlying molecular mechanism. If ERAP1 regulation of the Hh pathway were going through betaTrCP, one would not expect it to have much of an effect in Ptc mutant MEFs where Smo is active and the betaTrCP degron should not be phosphorylated. Yet, inhibition of ERAP1 has a very strong effect. I think it is important to demonstrate that the effects of ERAP1 depend upon the presence of the betaTrCP degron in the Gli proteins.

To address this question we tested the phosphorylation of Gli1 by PKA in Ptc^{-/-} MEFs, since the Gli factors are recognized by the SCF^{betaTrCP} ubiquitin ligase upon phosphorylation by PKA. To this end, we performed an immunoprecipitation assay in Ptc^{-/-} MEFs using an anti-Gli1 antibody followed by immunoblot analysis with an anti-PKA phospho-substrate in these cells. The experiment shown in the **new Supplementary Figure 4a** demonstrates that Gli1 is phosphorylated by PKA. Consistent with these data, the phosphorylation of Gli1 is strongly reduced following treatment with PKA inhibitor H89 resulting in an increased Gli stability as demonstrated by significant increase in protein levels (**new Supplementary Figure 4b**). Accordingly, whereas Gli1 protein is sensitive to ERAP1 activity, the Gli1 protein mutant deleted in the betaTrCP degron (Gli1ΔC) is not, thus confirming that the effect of ERAP1 depends upon the presence of the betaTrCP degron in the Gli proteins. These data are shown in the new **Figures 2k and 2l** and discussed in the manuscript (**page 6**).

2. The authors should also examine whether ERAP1 has any effects on SPOP as this E-3 ligase complex is also known to regulate the Gli proteins.

Thank you for the suggestion. The effect of ERAP1 modulation on the E3 ubiquitin ligase SPOP protein has been investigated. As shown in the revised **Figures 2d-f**, both ectopic expression and pharmacological or genetic inhibition of ERAP1 do not affect the levels of the SPOP protein. Data are shown in the new **Figures 2d-f** and discussed in the manuscript (**page 5**).

Reviewer #2, Expertise: Ubiquitin (Remarks to the Author):

Bufalieri and colleagues described a novel role for ERAP1 as a positive regulator of the Hedgehog signaling and as a promising target for the treatment of Hh-dependent tumors. The authors explored the molecular basis by which ERAP1 controls the Hh-pathway, focusing on the USP47-betaTrCP circuitry. They demonstrated that ERAP1 physically interacts with the betaTrCP-associated deubiquitylase enzyme USP47, thus, promoting betaTrCP degradation and, consequently, Gli factors stabilization. Based on these results, the authors evaluated the possibility to target ERAP1 to impair Hh-driven tumor growth. The inhibition of ERAP1 shows to have a significant impact on tumor growth both in vitro and in vivo. The central findings of this work are provocative and of solid, helping our understanding of the regulation of the Hh pathway and providing insights for the clinically implications of targeting Hh pathway. In particular, the study establishes an interesting and novel role for ERAP1 in the control of tumorigenesis, independently from its function in the immune system. Moreover, this study added valuable information about the involvement of upstream signals that control betaTrCP-mediated degradation of Gli1, an important nodal point in Hh signaling regulation. Overall, the experiments are of high quality and the paper is clearly written. This reviewer appreciates the novelty of this work and believes it would warrant publication after addressing the points raised below:

We thank the Reviewer for appreciating the importance and soundness of our data.

1. Figure 2a-c: the authors show the effects of either ERAP1 overexpression or its pharmacological inhibition on the protein levels of Gli factors. To make the results more homogeneous, the authors should show also in Figure 2f the protein levels of Gli2 and Gli3.

As requested by the Reviewer, the protein levels of Gli2 and Gli3 (both the full length and the cleaved form, Gli3FL and Gli3R, respectively) are included in the revised **Figure 2f** and discussed in the manuscript (**page 5**). Data in Figure 2f show that while genetic inhibition of ERAP1 decreases expression of Gli1 and Gli2 and increased those of Gli3FL and Gli3R and betaTrCP, the reintroduction of ERAP1 results in the opposite effect.

2. The authors shown that the modulation of ERAP1 affects protein levels and stability of β TrCP. In light of these data, it would be important analyze the effect of ERAP1 inhibition on other β TrCP substrates.

As requested by the Reviewer, the effect of ERAP1 inhibition has been investigated on other substrates, including GBF1, β -catenin, Cdc25a and IKB α . No significant change was observed for selected targets. Data are shown in the **Supplementary Figure 3c and 3d** and discussed in the manuscript (**page 5**).

3. Figure 3: the authors explain the molecular mechanism by which ERAP1 affects β TrCP stability (i.e., through the interaction with USP47). In Figure 6, the effect of ERAP on the Hh-dependent tumor growth was established by inhibiting ERAP1 both pharmacologically and genetically. Is the interaction between USP47 and β TrCP dependent on ERAP's activity? The authors should address this point by verifying the formation of β TrCP/USP47 complex following pharmacological inhibition of ERAP1 by Leu-SH treatment. Similarly to what shown in Figure 3l, one would expect an increase of the β TrCP/USP47 interaction after inhibition of ERAP1 activity.

We thank the Reviewer for raising this very important point. We addressed the question performing coimmunoprecipitation experiments in the presence of different amounts of Leu-SH. As shown in new Figure 3 m, the interaction between β TrCP and USP47 was significant increased following inhibition of ERAP1 activity. Data are shown in the revised **Figure 3m** and discussed in the manuscript (**page 7**).

4. The effect of USP47 knockdown on β TrCP protein levels should be investigated in USP47^{-/-} MEFs after reintroduction of USP47.

As requested by the Reviewer, the effect of USP47 knockdown on β TrCP protein levels has been investigated in USP47^{-/-} MEFs after reintroduction of USP47. Although, a total rescue in β TrCP protein levels was not observed due to low transfection efficacy of USP47^{-/-} MEFs, we observed increased levels of β TrCP in USP47^{-/-} MEFs after reintroduction of USP47. Data are shown in the **Figure 3g** and discussed in the manuscript (**page 6**).

5. Further, Gli protein levels should be added in Figure 3f.

As requested, Gli protein levels have been added in **Figure 3f** and discussed in the manuscript (**pag 6**).

6. The authors have provided several evidences of the effect of ERAP inhibition on Hh-dependent tumor growth, both in vitro and in vivo using medulloblastoma cells derived from Ptch^{+/-} mice. Does any medulloblastoma cell line behave similarly?

Similar to murine tumor models, both pharmacological and genetic inhibition of human ERAP1 affects cell proliferation and increases tumor cell death in Daoy, a well characterized human Hh-dependent medulloblastoma cell line (Triscott et al., Cancer Res 73:6734, 2013; Ivanov et al., JBiotechnology 236:10, 2016; Northcott et al., Acta Neuropathol 123:615, 2012). Data are shown in **Supplementary Figure 7** and discussed in the manuscript (**pag 10**).

7. Since the strong results obtained in SHH-MB Patient-Derived Xenograft culture, the authors might want to verify the effect of ERAP1 modulation on medulloblastoma PDX growth also in vivo.

Thank you for the suggestion. The pharmacological and genetic inhibition of ERAP1 reduces cell growth and promotes apoptosis in SHH-MB PDX also in vivo. Data are shown in **Figure 7 f-k** and **Supplementary Figure 8** and discussed in the manuscript (**pag 10**).

Reviewer #3, Expertise: Medulloblastoma (Remarks to the Author):

Bufalieri et al. present a manuscript where ERAP1 is shown to promote HH-dependent brain tumorigenesis by controlling USP47-mediated degradation of U3 ligase, Beta-TrCP. They show that ERAP1 physically binds to USP47 and thereby promoting Beta-TrCP degradation that normally control Gli protein stability. They further suggest using ERAP1 inhibitors to suppress SHH tumor formation and present this both in vitro and in vivo. The manuscript is well-written and experiments are solid. I still have a few concerns about the conclusions made from the obtained results.

We thank the Reviewer for the positive general comments on our study.

Major comments:

Reviewer: What was the initial reason to study ERAP1 in HH tumors in particular in the first place? This is not clearly motivated in the introduction section.

We thank the Reviewer for raising this important point that needs clarification. While exploring novel compounds as Hh inhibitors, we have screened a library of small molecule and we found that leucinethiol (Leu-SH), a potent inhibitor of ERAP1, strongly reduced the Hh/Gli1 activity. We were intrigued by this new potential role of ERAP1 that we decided to study in detail. This point is now discussed in the revised manuscript (**pag. 3**).

2. **Reviewer:** Although the shERAP1 construct used in the manuscript shows great downregulation of ERAP1 as compared to shControl, it would still be good to test at least one additional shERAP1 construct to confirm similar downregulation and further rule out off target effects in Fig 1.

As requested by the Reviewer, the effect of ERAP1 inhibition has been confirmed by an additional shERAP1 construct. Data are shown in **Figures 1b, 1c, 1g and 1h** and discussed in the manuscript (**pages 4 and 5**).

3. **Reviewer:** In Fig. 1d-h it is not clear how the expression levels are normalized and to what.

As requested, this point has been clarified in the method section (**pag 16**).

4. **Reviewer:** Experiments and results in Figs 2-4 are well performed and mechanistically relevant. However, it is not clear how specific Leu-SH is in targeting ERAP1 as compared to other aminopeptidases. A blot for other APs would have been explanatory. For example, only the highest concentrations 10-30uM of Leu-SH are regulating ERAP1 itself. Protein levels are not reduced in 1uM concentrations perhaps suggesting that the drug is not that specific.

Leu-SH is a potent inhibitor of the ERAP1 activity. It was originally designed as an inhibitor of leucine aminopeptidase by analogy with sulphhydryl inhibitors of other zinc-containing peptidases (Chan WW. 1983). Currently it is the only known compound to efficiently inhibit ERAP1 in cellular models, both *in vitro* and *in vivo* as documented (Serwold T et al., Nat Immunol. 2001; Saric T et al., Nat Immunol. 2002; Serwold T et al., Nature 2002; Guil S et al., J Biol Chem. 2006; Kanaseki T et al., Immunity 2006; Hammer GE et al., Nature Immunology 2007; Parmentier N et al., Nat Immunol. 2010; Nagarajan NA et al., Nature Immunology 2012; James E et al., J Immunol 2013; Cifaldi L et al., Cancer Res. 2015; Dasari V et al., Immunol Cell Biol 2016; Ma W et al., J Immunol. 2016; Ma W et al., J Immunol. 2019). The optimal concentration is in the range of 10-30 μ M. Leu-SH binds to the active site of the enzyme without affecting protein expression and/or stability. The small variations in protein expression detected at the higher concentrations are not significant (as shown by the densitometric analysis of three independent experiments in Supplementary Figure 2a). However, as discussed below (Minor comment 2), our intent is not to propose Leu-SH as a drug to cure MB, but to highlight the role of ERAP1 in the Hh signaling pathway and propose its inhibition as a promising therapeutic target for Hh-driven tumors. In this regard, the effects obtained by the pharmacological inhibition of ERAP1 were confirmed with experiments of genetic depletion of ERAP1.

5. **Reviewer:** In Fig 3, USP47 is picked and suggested as a DUB that would regulate Beta-TrCP. However other potential DUBs, e.g. USP24 (Wang et al. Nat Comm., 2018) that might regulate Beta-TrCP were not investigated, why?

We thank the reviewer for raising this interesting point. In the previous version of the manuscript it was not discussed because we were not aware of the Wang's data unpublished at the time. The potential involvement of other DUBs that might control β TrCP activity, such as USP24 and USP22, has been investigated. To this end, we carried out co-IP assay and we did not observed interaction between ERAP1 and USP24 or USP22. Data are shown in **Supplemental Figure 5c** and discussed in the revised manuscript (**pag 6**).

6. **Reviewer:** In Fig 5b authors describe how cell death is increased but not what type of cell death (apoptosis, senescence, necrosis, cell cycle arrest, etc.).

Thank you for the suggestion. Additional analyses evaluating how cell death is increased by Leu-SH in the primary MB cells have been performed. Treatment with ERAP1 inhibitor leads to apoptosis of primary MB cells and SHH-PDX, as indicated by the increased levels of cleaved caspase-3. Data are shown in **Figure 5c, Figure 7 h-i-k**, and discussed in the manuscript (**pages 8 and 10**).

7. **Reviewer:** Authors show significant reduction in cell growth in Fig 5j as a function of confluency over time. However, image-based growth assessment has several disadvantages such as local differences in density etc. Cell number counting, impedance or cell-metabolism assays might be more appropriate in this case.

As rightly pointed out by the Reviewer, we used only an image-based growth assessment to show the reduction in cell growth. The revised version has been implemented with more appropriate experiments, including the cell number counting and a cell-metabolism assay. New data are shown in **Figure 7 a-c** and discussed in the manuscript (**page 10**).

8. **Reviewer:** In Fig 5k authors demonstrate late apoptotic events by PI staining. Annexin 5 co-staining and FACS analysis might supplement the data in the better way to detect early apoptotic events at lower concentrations.

As requested by the Reviewer, Annexin V co-staining and FACS analysis were performed to evaluate the early apoptotic events on the SHH-MB Patient-Derived (ICN MB PDX 12) cells. Unfortunately, we did not get any result because the

cells being very sensitive to detachment or harvesting were found to be positive for Annexin V staining, even if untreated. We overcame the problem by evaluating the percentage of Caspase-3 positive cells following treatment with Leu-SH inhibitor. Data are shown in **Figure 7d and 7e** and discussed in the manuscript (**page 10**).

9. **Reviewer:** In Fig 6c authors indicate reduced proliferation and increased cellularity of the tumors. This is a very promising finding, however does not indicate by which means the effect was achieved. Apoptosis marker analysis e.g. Cl. Cas 3 staining in combination with staining for neuronal differentiation markers such as SYP or NeuN shall be investigated.

As requested by the Reviewer, cleaved Caspase-3 and NeuN proteins were studied by both immunohistochemistry and immunoblot analyses. Data shown in **Figures 6c, 6d and 6f** and discussed in the manuscript (**pages 8 and 9**) indicate that ERAP1 inhibition reduces MB growth by blocking cell proliferation and promoting both apoptosis and differentiation.

10. **Reviewer:** The authors use in vivo allografts from a mouse SHH MB model harbouring Ptc1 mutation. In vivo experiments with a flank tumor that does not include a natural tumor niche for MB growth. A mouse model that spontaneously develops SHH MB or orthotopic transplants of the tumor would have been more relevant for the pharmacological studies. In addition, a survival curve of the treated mice would be beneficial for the significance of their data.

To address the points raised by the Reviewer we performed the following experiments:

- i) we determined if Leu-SH was also effective on spontaneously derived intracranial SHH-MB, Gfap-Cre/Ptc^{fl/fl} mice showing MB at neonatal stage. To this purpose, Hh targeted genes and β TrCP expression was evaluated by qRT-PCR and immunoblot analyses in MB recovered from Gfap-Cre/Ptc^{fl/fl} mice treated with Leu-SH or vehicle for two days. The data obtained indicate that pharmacological inhibition of ERAP1 represses Hh-mediated transcription, reducing Gli1 and increasing β TrCP protein levels (see **Supplementary Figures 6a and 6b** and **page 9**).
- ii) We carried out pharmacological studies also in an orthotopic allograft model. To this end, primary MB cells isolated from Math1-cre/Ptc^{C/C} mice tumors were implanted into the cerebellum of NSG mice. The animals treated with Leu-SH showed a significant reduction of tumor masses as compared to control (see **Figure 6g,h** and **page 9**).
- iii) To study if Leu-SH prolongs the survival of SHH-MB-prone mice, Math1-Cre/Ptc^{fl/fl} mice were treated with i.p. administered Leu-SH, starting from 4 weeks of age. The average survival was significantly increased in Leu-SH-treated mice compared to controls, demonstrating the therapeutic benefit of this drug in tumors developed *in situ*. Data are shown in **Figure 6q** and discussed in the manuscript (**page 9**).

11. **Reviewer:** In order to increase the clinical relevance of the story, the manuscript would benefit from checking levels of USP47 and ERAP1 and see if they are differentially expressed in SHH MB patients as compared to other molecular subgroups of MB tumors or other brain tumor entities. Such data is readily and publically available.

As suggested by the Reviewer, the levels of USP47 and ERAP1 were evaluated in SHH-MB patients and compared with the other molecular subgroups. No difference was observed between the different subgroups. Data are shown in the **Supplementary Figure 9** and discussed in the manuscript (**page 12**).

Minor comments

1. **Reviewer:** In Supplementary Fig. 1b it seems like there is a trend in increasing luc activity. I would suggest further increase in Leu-SH to confirm the hypothesis that the ERAP1 inhibition does not affect the beta-Catenin pathway.

We tried to use higher concentrations of Leu-SH, but it was toxic to cells. However, although there is a trend this is not significant.

2. **Reviewer:** From a clinical perspective, ERAP inhibition will probably lead to immune escape suggesting it would be a bad target for certain cancers and for tumor recurrence. Further, if aminopeptidase inhibitors like Leu-SH at all would be potential drug candidates in children is not clear. Any of these potential limitations should be included in the discussion.

We thank the Reviewer for raising this point. The activity of ERAP1 has been shown to be important in shaping the final repertoire of peptides presented to T cells. In murine models, ERAP1 has been shown to be important in the immune responses to certain viruses and bacteria. The seminal work by Shastri group on mice lacking ERAP1 (Serwold et al Nature 2002, Hammer et al. Nat Immunol 2006, Hammer et al. Nat Immunol 2007), demonstrated that ERAP1 inhibition results in a modest reduction of MHC class I surface expression, as compared to those deriving from the loss of the other antigen processing components (i.e., TAP, LMP2, LMP7 and β 2m). They demonstrate that immunization of ERAP1-deficient mice with splenocytes from wild-type mice, and vice versa, of wild-type mice with splenocytes from ERAP1-deficient mice, resulted in potent CD8⁺ T cell responses, thus suggesting that the lack of ERAP1 alters the normal repertoire of peptides bound to MHC class I and, consequently, the CD8⁺ T cells responses in mice.

Moreover, recent genome-wide studies have strongly associated ERAP1 polymorphisms with ankylosing spondylitis and other autoimmune diseases (Fruci et al. Tissue Antigens 2014, Stratikos et al. Front Oncol. 2014). The polymorphic residues map to ERAP1's catalytic and regulatory sites and alter peptide specificity and processing activity, thus suggesting that the enzymatic activity of ERAP1 is important in the link with genetic disease. However, the lack of highly specific chemical compounds for ERAP1 has constrained the progress in this area. Given the great interest in ERAP1,

many companies are investing in the development of ERAP1 inhibitors for potential therapeutic intervention. As requested by the Reviewer, these points are discussed in the revised version of the manuscript (**page 11**).

3. **Reviewer:** Figure 4 and Supplementary Figure 4 are exactly the same.

Sorry for this mistake. The correct figures have been uploaded.

4. **Reviewer:** In Fig5 the effect of ERAP1 inhibition on Oct4 are remarkable and do not follow pharmacological patten of other targets that were tested. Further follow-up on possible of target regulation or commenting on the explanation of this would be appropriate.

Thank you for the suggestion. Additional studies will be performed to investigate the potential role of ERAP1 in the stability of Oct4 at protein levels. Of course we cannot exclude that other pathways may converge with ERAP1-mediated pro-tumorigenic effect.

5. **Reviewer:** In the last section the heading is wrong: "ERAP1 inhibits Hh-dependent tumor growth in vivo". The heading has been reformulated (**page 8**).

REVIEWERS' COMMENTS:

Reviewer #1 (Remarks to the Author):

The authors have nicely addressed the two concerns that I had with the paper. They have also made substantial additions to the paper to address the concerns of the other two reviewers.

Reviewer #2 (Remarks to the Author):

I have examined the revised manuscript. The new experiments in the paper have completely satisfied my previous concerns. The paper is a wonderfully clear and concise description of many interesting findings, and it is clearly suitable for publication.

Reviewer #3 (Remarks to the Author):

Authors answered all of my questions and concerns. I have nothing further to address.